# COMBINATORIAL BANDITS FOR MAXIMUM VALUE REWARD FUNCTION UNDER VALUE-INDEX FEEDBACK

**Yiliu Wang**[*]
Allen Institute
Seattle, WA 98109, USA
`yiliu.wang@alleninstitute.org`

**Wei Chen**
Microsoft Research
Beijing, China
`weic@microsoft.com`

**Milan Vojnović**
Department of Statistics, LSE
London, United Kingdom
`m.vojnovic@lse.ac.uk`

## ABSTRACT

We investigate the combinatorial multi-armed bandit problem where an action is to select $k$ arms from a set of base arms, and its reward is the maximum of the sample values of these $k$ arms, under a weak feedback structure that only returns the value and index of the arm with the maximum value. This novel feedback structure is much weaker than the semi-bandit feedback previously studied and is only slightly stronger than the full-bandit feedback, and thus it presents a new challenge for the online learning task. We propose an algorithm and derive a regret bound for instances where arm outcomes follow distributions with finite supports. Our algorithm introduces a novel concept of biased arm replacement to address the weak feedback challenge, and it achieves a distribution-dependent regret bound of $O((nk/\Delta)\log(T))$ and a distribution-independent regret bound of $\tilde{O}(\sqrt{nkT})$, where $\Delta$ is the reward gap and $T$ is the time horizon. Notably, our regret bound is comparable to the bounds obtained under the more informative semi-bandit feedback. We demonstrate the effectiveness of our algorithm through experimental results.

## 1 INTRODUCTION

We address a combinatorial bandit problem where, in each round, the agent selects a set of $k$ arms from a pool of $n$ arms. Each selected arm produces an independent random outcome, and the reward of the selected set of arms is determined by the maximum value among these outcomes. Following the selection of arms in a round, the agent receives feedback consisting of the maximum outcome value and the identity of the arm that achieved this value. The goal of the agent is to maximize the reward accrued over a time horizon. We term this sequential decision-making problem the $k$-MAX *bandit with max value-index feedback*. The outcomes of arms are assumed to be independent random variables across arms and rounds. Initially, we consider arm outcomes according to binary distributions, and then extend our analysis to arbitrary discrete distributions with finite supports. The performance of the agent is evaluated based on the expected cumulative regret over a predefined time horizon, which quantifies the difference between the cumulative reward achieved by selecting a set with maximum expected reward in each round and the cumulative reward achieved by the agent.

As a concrete motivating application scenario, consider an online platform recommending some products, e.g., Netflix, Amazon, and Spotify. Assume there is in total $n$ products for recommendation on the platform. The goal is to display a subset of $k$ product items to a user that best fit the user preference. The feedback can be limited, as we may only observe which displayed product item the user clicked on and their subsequent rating of that product. Many other problems can be formulated in our setting, such as project portfolio selection (Blau et al., 2004; Jekunen, 2014), team formation

---

[*]The work was conducted during an internship at Microsoft Research and as part of a thesis at the London School of Economics.

(Kleinberg & Raghu, 2018; Sekar et al., 2021; Lee et al., 2023; Mehta et al., 2020), and sensor placement problems (Golovin & Krause, 2011; Asadpour & Nazerzadeh, 2016).

The problem presents two main challenges. Firstly, the reward function is the maximum value function, which is nonlinear and depends not only on the expected values of the constituent base arms. As demonstrated in the numerical section, high-risk, high-reward arms may outperform stable-outcome arms in this scenario, even though their expected outcome may be the same. The second challenge is due to the limited feedback. The agent only observes the maximum value and the identity of an arm that achieves this value, which makes it difficult to learn about other non-winning arms.

The problem we consider shares similarities with some combinatorial bandit problems (Cesa-Bianchi & Lugosi, 2012; Chen et al., 2013; 2016b), but it differs in the feedback structure. Specifically, the max value-index feedback structure is neither the semi-bandit nor the full-bandit feedback, which are commonly studied in the bandit literature. To elaborate, let $X_1, \ldots, X_n$ be independent random variables corresponding to arm outcomes. In the semi-bandit feedback, the feedback consists of all values $\{X_i \mid i \in S\}$ for any set $S$ of arms. In contrast, the full-bandit feedback includes only the maximum value, $\max\{X_i \mid i \in S\}$, under the maximum value reward function. The max value-index feedback, on the other hand, provides both the maximum value $\max\{X_i \mid i \in S\}$ and the index $I \in \arg\max\{X_i \mid i \in S\}$. Thus, the max value-index feedback is much weaker than the semi-bandit feedback, and only slightly stronger than the full-bandit feedback. This week feedback structure provides very limited information on a non-winning arm $j \neq I$: it is unclear whether $j$ loses because its current outcome is low by chance, or it cannot win anyway even with its highest support value. This poses a severe challenge in learning the information of all arms.

We present algorithms and regret bounds for the $k$-MAX problem with max value-index feedback, under different assumptions regarding the information available about arm outcomes. Our results show that despite considerably more limited feedback, comparable regret bounds to those of combinatorial semi-bandits can be achieved for the $k$-MAX bandit problem with max value-index feedback.

## 1.1 RELATED WORK

The problem we study has connections with combinatorial multi-armed bandits (CMAB) (Cesa-Bianchi & Lugosi, 2012; Chen et al., 2013; 2016b). Most existing work on CMAB problems focuses on the semi-bandit feedback setting, as seen in (Chen et al., 2013; Kveton et al., 2015a). The $k$-MAX problem with semi-bandit feedback was addressed in (Chen et al., 2016a), where the solution is simpler compared to our paper due to the semi-bandit feedback assumption.

In most works on full-bandit CMAB problems, restrictions are typically imposed on the reward function. For instance, Rejwan & Mansour (2020) considered the sum reward function, while only a few algorithms have been proposed for non-linear reward functions. In a bipartite setting where the agent selects a pair of arms from a row of $K$ items and a column of $L$ items, Katariya et al. (2017) examined the minimum value reward function for rank-1 bandit problems. It is worth noting that the rank-1 assumption holds for Bernoulli-distributed arms in the general setting under the maximum value reward function. On the other hand, Gopalan et al. (2014) investigated the full-bandit CMAB with general rewards using a Thompson sampling algorithm. However, computing the posteriors in the algorithm is computationally intensive, and the regret bound has a large exponential constant. More recently, Agarwal et al. (2021a) proposed a merge and sort algorithm under the assumption that the distributions of arm outcomes obey a first-order stochastic dominance (FSD) condition. However, this condition is restrictive, as it fails to hold for binary distributions. Additionally, Agarwal et al. (2021b) proposed an adaptive accept-reject algorithm (DART) that sequentially identifies good arms based on mean rewards obtained by playing actions containing that arm. However, this algorithm relies on the arm-ordering assumption that the goodness of arms is ordered by their expected reward, which is not satisfied in $k$-MAX bandit. In summary, existing full-bandit CMAB solutions either do not apply to our problem or have exponential computational complexity.

A related work is on combinatorial cascading bandits, e.g. Kveton et al. (2015b). Here the agent chooses an ordered sequence from the set of arms and the outcomes of arms are revealed one by one until a stopping criteria is met. Chen et al. (2016b) generalized the problem to combinatorial semi-bandits with probabilistically triggered arms (CMAB-T). The main difference with our setting is that CMAB-T and cascading bandits in particular assume more feedback information and is inherently semi-bandit. In the recommendation system scenario, cascading bandits assume that the user would follow the recommendation order and investigate each item one by one and stop when clicking on an

Table 1: Known regret bounds for CMAB problems under different settings.

| | Feedback | Reward | Assumptions | Regret |
|---|---|---|---|---|
| Chen et al. (2016a) | semi-bandit | general | general | $O(\frac{nk}{\Delta}\log(T))$ |
| Rejwan & Mansour (2020) | full-bandit | linear | sub-Gaussian | $O(\frac{nk}{\Delta}\log(T))$ |
| Katariya et al. (2017) | full-bandit | min | rank-1 | $O(\frac{K+L}{\Delta}\log(T))$ |
| Agarwal et al. (2021a) | full-bandit | general | FSD | $\tilde{O}(n^{1/3}k^{1/2}T^{2/3})$ |
| Agarwal et al. (2021b) | full-bandit | general | arm-ordering | $\tilde{O}(k\sqrt{nkT})$ |
| Kveton et al. (2015b) | cascading-bandit | max | Bernoulli | $O(\frac{nk}{\Delta}\log(T))$ |
| Our work | value-index | max | general | $O(\frac{nk}{\Delta}\log(T))$ |

attractive item, while our setting essentially removes this sequential order assumption and allows the user to browse through the items in any order and only returns the value and index of the selected item as the feedback. Another difference with the cascading bandit is that we consider more general distributions of arm outcomes. Our work also bears resemblance to MNL bandits (Agrawal et al., 2019), in which the user selects an item following a multinormial logit (MNL) model, which is different from our $k$-MAX selection model. They assume that item values are known, while handling unknown item values is one of the main challenges we are facing.

We summarize some known results on regret bounds for CMAB problems in Table 1. In the table, $\Delta$ denotes the gap between the optimum expected regret of a set and the best suboptimal expected regret of a set. Agarwal et al. (2021a) and Agarwal et al. (2021b) only provide distribution-independent regret bounds, which are worse than the $\tilde{O}(\sqrt{nkT})$ distribution-independent regret bounds that follow from our distribution-dependent regret bounds.

The index feedback bear resemblance to comparison observations in dueling bandits (Ailon et al., 2014; Sui et al., 2017), which only provide which arm wins in a two-way or multi-way comparisons, with the purpose of identifying the single best arm. In contrast, our approach incorporates the value information to guide the learning process with the aim of finding the best combination of arms with the expected maximum value outcome.

## 1.2 SUMMARY OF CONTRIBUTIONS

Our contributions can be summarized as follows:

• We introduce and investigate a novel combinatorial bandit problem, which involves maximizing the reward function based on maximum value-index feedback. This feedback structure falls between the extensively studied full-bandit and semi-bandit feedback types, offering slightly more information than full-bandit feedback. In contrast to the full-bandit scenario, we incorporate additional knowledge of the maximum-value index, a feature observed in certain real-world applications. Our work can be viewed as a progression towards resolving full-bandit CMAB problems with non-linear reward functions, under relatively modest assumptions.

• We introduce algorithms tailored for binary distributions of arm outcomes. We first discuss the case when the order of arm outcome values is known. Then for the case when the ordering of values is a priori unknown to the algorithm, we propose a variant of the CUCB algorithm that leverages the novel concept of biased arm replacement. We demonstrate that this algorithm achieves a regret upper bound comparable to that observed in scenarios where the ordering of values is known in advance. Consequently, it offers performance similar to that achievable under semi-bandit feedback.

• We extend the algorithm to address the more general case of arbitrary discrete distributions with finite supports. By establishing an equivalence relationship between the binary distributions and arbitrary discrete distributions, we reduce the general case back to the binary distribution case to solve the problem.

**Organization of the paper** In Section 2, we formally define the problem. In Section 3, we first establish key properties of the reward function and then present our algorithms and regret bounds for the case of binary distributions of arm outcomes. Section 4 discusses extensions to arbitrary discrete distributions with finite supports. Our numerical results are presented in Section 5. Finally, we summarize our work in Section 6. Proofs of theorems are provided in the Appendix.

## 2 PROBLEM FORMULATION

We consider a sequential decision-making problem involving an agent and a set of $n$ arms, denoted as $\mathcal{A} = [n] = \{1, 2, \ldots, n\}$. For each arm $i \in \mathcal{A}$, its outcome is sampled according to a random variable $X_i$ with a discrete distribution having a finite support. The outcomes of arms in all rounds are independent across both arms and rounds. Let $0 = v_{i,0} < v_{i,1} < \cdots < v_{i,s_i}$ denote the values of the support of the distribution of $X_i$, where $s_i$ is a positive integer, and $s_i + 1$ is the size of the support. Let $p_{i,j} = \Pr[X_i = v_{i,j}]$ for $j \in \{0, 1, \ldots, s_i\}$, with $0 < \sum_{j=1}^{s_i} p_{i,j} \leq 1$. Let $\boldsymbol{v} = (\boldsymbol{v}_1, \ldots, \boldsymbol{v}_n)$ and $\boldsymbol{p} = (\boldsymbol{p}_1, \ldots, \boldsymbol{p}_n)$, where $\boldsymbol{v}_i = (v_{i,1}, \ldots, v_{i,s_i})$ with $v_{i,j} \in [0, 1]$, and $\boldsymbol{p}_i = (p_{i,1}, \ldots, p_{i,s_i})$. For the special case of binary distributions, we use $p_i$ and $v_i$ instead of $p_{i,1}$ and $v_{i,1}$, respectively. Both $\boldsymbol{v}$ and $\boldsymbol{p}$, as well as the $s_i$'s in the general case, are unknown parameters to the agent. Further discussion about the real-world implications of our assumptions regarding the arm support is provided in Appendix A.

We define $\mathcal{F} = \{S \in 2^{\mathcal{A}} \mid |S| = k\}$ as the set of actions, where $0 < k < n$ is an integer. At each round $t$, the agent selects an action $S_t \in \mathcal{F}$. The agent observes the maximum outcome value of selected arms and the index of an arm achieving this maximum outcome value, and receives the reward corresponding to the maximum outcome value. We denote the expected reward of an action $S$ as $r_S(\boldsymbol{p}, \boldsymbol{v}) = \mathbb{E}[\max\{X_i \mid i \in S\}]$, which is a function of parameters $\boldsymbol{p}$ and $\boldsymbol{v}$.

The performance of the agent is measured by the cumulative regret, defined as the difference of the expected cumulative reward achieved by playing the best action and the expected cumulative reward achieved by the agent. Denote $\mathrm{OPT}(p, v) = \max\{r_S(p, v) \mid S \in \mathcal{F}\}$. An $(\alpha, \beta)$-approximation oracle takes $(p', v')$ as input and returns a set $S$ such that $\Pr[r_S(p', v') \geq \alpha \, \mathrm{OPT}(p', v')] \geq \beta$ where $\alpha$ is the approximation ratio and $\beta$ is the success probability. If the agent uses an $(\alpha, \beta)$-approximation oracle, then we consider the $(\alpha, \beta)$-approximation regret defined as

$$R(T) = T \, \alpha \, \beta \, \mathrm{OPT}(p, v) - \mathbb{E}\left[\sum_{t=1}^{T} r_{S_t}(p, v)\right].$$

The general offline $k$-MAX problem can be approximately solved by a greedy algorithm to achieve a $(1 - 1/e)$ approximation, or by a polynomial-time approximation scheme (PTAS) to achieve a $(1 - \varepsilon)$ approximation for any $\varepsilon > 0$ (Chen et al., 2016a). The binary $k$-MAX problem can be solved exactly using dynamic programming (Du et al., 2024), and the Bernoulli $k$-MAX problem can be solved exactly by ordering values and selecting the top $k$ valued items.

## 3 ALGORITHMS AND REGRET BOUNDS FOR BINARY DISTRIBUTIONS

In this section, we present algorithms and regret bounds for the $k$-MAX problem with max value-index feedback with arm outcomes according to binary distributions. We first establish some properties of reward functions which are crucial for our regret analysis. Next we discuss the scenario when the ordering of $v_1, \ldots, v_n$ values is known. Then, we consider the case when the ordering of $v_1, \ldots, v_n$ is a priori unknown. We present an algorithm and establish that it achieves the same regret bound as when the ordering is known, up to constant factors.

For the convenience of exposition, we assume that values $v_1, \ldots, v_n$ are distinct. This ensures that for any action $S_t$, there is a unique arm achieving the maximum value over the arms in $S_t$. Alternatively, allowing for non-unique values can be addressed by using a deterministic tie-breaking rule. Further discussion on this matter can be found in Appendix B.

We define an extended set $\mathcal{B}$ of arms as the union of two sets of arms, namely $\mathcal{Z}$ and $\mathcal{V}$. The first set $\mathcal{Z} = a_1^z, \ldots, a_n^z$ consists of $n$ arms whose outcomes are according to independent Bernoulli random variables $Z_1, \ldots, Z_n$ with mean values $p_1, \ldots, p_n$. The second set $\mathcal{V} = \{a_1^v, \ldots, a_n^v\}$ consists of $n$ arms whose outcomes are deterministic with corresponding values $v_1, \ldots, v_n$. Note that the outcome of each arm $i \in \mathcal{A}$ can be written as $X_i = V_i Z_i$. Each time an action $S_t$ is played, we obtain information for some of the arms in $\mathcal{B}$. We call these arms *triggered arms* and observe their outcome values as feedback. An important observation is that when arms are ordered in decreasing order with respect to their values, lower-order arms being triggered implies that higher-order arms had zero outcomes. We define $T_{i,t}^z$ as the number of times arm $a_i^z$ is triggered, and $T_{i,t}^v$ as the number of times arm $a_i^v$ is triggered up to time step $t$.

**Additional notation** For any two vectors $\boldsymbol{x}, \boldsymbol{x}' \in \mathbb{R}^n$, we write $\boldsymbol{x} \geq \boldsymbol{x}'$ if $x_i \geq x_i'$ for all $i \in [n]$.

### 3.1 PROPERTIES OF REWARD FUNCTION

For the prevailing case of binary distributions of arm outcomes, for any set $S \in \mathcal{F}$, under the assumption that arms in $\mathcal{A}$ are ordered in decreasing order of values $v_1, \ldots, v_n$, the expected reward can be expressed as:

$$r_S(\boldsymbol{p}, \boldsymbol{v}) = \sum_{i \in S} v_i p_i \prod_{j \in S: j < i} (1 - p_j), \tag{3.1}$$

where the product over an empty set has a value of 1.

There are two key properties of the reward function in (3.1) that we leverage in our regret analysis: monotonicity and bounded smoothness, as we define and reason below.

**Monotonicity** The first property is monotonicity.

**Lemma 3.1.** *For every $S \in \mathcal{F}$, $r_S(\boldsymbol{p}, \boldsymbol{v})$ is increasing in every $p_i$ and $v_i$.*

It is clear from (3.1) that $r_S(\boldsymbol{p}, \boldsymbol{v})$ is monotonically increasing in $v_i$. Additionally, it can be shown to be monotonically increasing in $p_i$ by taking the first-order derivative with respect to $p_i$ and demonstrating that it is non-negative.

**Bounded Smoothness** The second property is *relative triggering probability modulated* (RTPM) bounded smoothness, which, with a slight abuse of notation, is defined as follows: for an arbitrary set of arms $\mathcal{B}$ with expected outcome values $\boldsymbol{\mu}$, and $r_S(\boldsymbol{\mu})$ denoting the expected reward of action $S$.

**Definition 3.2** (RTPM smoothness). The RTPM smoothness holds for reward functions $r_S$ if there exist constants $b_i > 0$ for $i \in [n]$ such that, for any two vectors of expected outcomes $\boldsymbol{\mu}$ and $\boldsymbol{\mu}'$, for every action set $S$,

$$|r_S(\boldsymbol{\mu}) - r_S(\boldsymbol{\mu}')| \leq \sum_{i \in S} q_i^{\boldsymbol{\mu}, S} b_i |\mu_i - \mu_i'|,$$

where $q_i^{\boldsymbol{\mu}, S}$ denotes the triggering probability of arm $i$ under action $S$ and the expected arm outcomes $\boldsymbol{\mu}$. Note that when $r_S(\boldsymbol{\mu})$ is increasing in $\boldsymbol{\mu}$, and $\boldsymbol{\mu} \geq \boldsymbol{\mu}'$, we can remove the absolute value in the above inequality.

The definition of RTPM smoothness is a generalization of the condition in (Wang & Chen, 2017), extended to incorporate the triggering probability $q_i^{\boldsymbol{\mu}, S}$ and weight $b_i$ to modulate the standard 1-norm condition. This modification allows us to account for arm-specific values, as we will demonstrate in our next lemma. The intuition behind this definition is that we underweight the importance of arms with small triggering probability in the expected reward, since even if we cannot accurately estimate the expected value of such arms, we incur very little loss in the expected reward due to their low triggering probabilities. This will be an important point in our regret analysis.

Let $q_i^{\boldsymbol{p}, S}$ be the triggering probability for arm $a_i^z$ and action $S$, and $\tilde{q}_i^{\boldsymbol{p}, S}$ be the triggering probability for arm $a_i^v$ and action $S$. Note that $q_i^{\boldsymbol{p}, S} = \prod_{j < i} (1 - p_j)$ and $\tilde{q}_i^{\boldsymbol{p}, S} = q_i^{\boldsymbol{p}, S} p_i$, because arm $a_i^z$ is triggered when all arms $j < i$ (with higher values) has outcome $Z_j = 0$, while arm $a_i^v$ is triggered when arm $i \in \mathcal{A}$ is the winning arm. The following lemma is a key for our regret analysis.

**Lemma 3.3.** *The expected reward functions in (3.1) satisfy the RTPM condition with respect to the extended set $\mathcal{B}$ of arms, i.e. for every $S$, $\boldsymbol{p}$ and $\boldsymbol{p}'$, $\boldsymbol{v}$, and $\boldsymbol{v}'$, it holds*

$$|r_S(\boldsymbol{p}, \boldsymbol{v}) - r_S(\boldsymbol{p}', \boldsymbol{v}')| \leq 2 \sum_{i \in S} q_i^{\boldsymbol{p}, S} v_i' |p_i - p_i'| + \sum_{i \in S} \tilde{q}_i^{\boldsymbol{p}, S} |v_i - v_i'|.$$

*Furthermore, if $\boldsymbol{p} \geq \boldsymbol{p}'$, then we can remove the factor 2 in the last inequality.*

The lemma can be intuitively explained as follows: When an arm $i \in \mathcal{A}$ has a small value $v_i'$ or the corresponding arm $a_i^z$ is unlikely to be triggered (small $q$), the estimation error on $p_i$ does not effect much on the reward estimate. Similarly, if the arm is unlikely to win (small $\tilde{q}$), the estimate error on $v_i$ does not affect much on the reward estimate. This concept is crucial for proving our regret bounds. For arms less likely to win due to small value or small winning probability, we may not be able to estimate their value and probability parameters accurately due to scarce observations. The lemma suggests that this is not a critical issue for regret analysis. To prove Lemma 3.3, we consider a sequence of vectors changing from $(\boldsymbol{p}, \boldsymbol{v})$ to $(\boldsymbol{p}', \boldsymbol{v}')$ and add up the changes in expected rewards. The full proof is provided in Appendix E.2.

---

**Algorithm 3.1** CUCB algorithm for unknown ordering of values

1: For $i \in \mathcal{A}$, $T_i^z \leftarrow 0, T_i^v \leftarrow 0$           ▷ Number of triggering times for $a_i^z$ and $a_i^v$
2: For $i \in \mathcal{A}$, $\hat{p}_i \leftarrow 1, \hat{v}_i \leftarrow 1$                  ▷ Initial estimation of parameters
3: **for** $t = 1, 2, \ldots, T$ **do**
4:      For $i \in \mathcal{A}$, $\rho_i \leftarrow \sqrt{\frac{3 \log(t)}{2 T_i}}, \tilde{\rho}_i \leftarrow \mathbf{1}\{T_i^v = 0\}$
5:      For $i \in \mathcal{A}$, $\bar{p}_i \leftarrow \min\{\hat{p}_i + \rho_i, 1\}, \bar{v}_i \leftarrow \min\{\hat{v}_i + \tilde{\rho}_i, 1\}$      ▷ UCB indices
6:      $S \leftarrow \text{Oracle}(\bar{\boldsymbol{p}}, \bar{\boldsymbol{v}})$      ▷ Oracle is the dynamic programming algorithm in (Du et al., 2024)
7:      Play $S$ and observe winner index $i^*$ and value $v_{i^*}$
8:      **if** $T_{i^*}^v = 0$ **then**
9:          Reset $T_{i^*}^z \leftarrow 0, \quad T_{i^*}^v \leftarrow 1, \quad \hat{v}_{i^*} \leftarrow v_{i^*}$
10:      **end if**
11:      For $i \in S$ such that $\hat{v}_i \geq v_{i^*}$: $T_i^z \leftarrow T_i^z + 1$
12:      For $i \in S$ such that $\hat{v}_i > v_{i^*}$: $\hat{p}_i \leftarrow (1 - 1/T_i^z)\hat{p}_i$
13:      For $i \in S$ such that $\hat{v}_i = v_{i^*}$: $\hat{p}_i \leftarrow (1 - 1/T_i^z)\hat{p}_i + 1/T_i^z$
14: **end for**

---

## 3.2 ALGORITHM FOR KNOWN ORDERING OF VALUES

We discuss the case of known ordering of values as a theoretically important step towards the more challenging case when the ordering of values is a priori unknown to the algorithm. For the case of known ordering of values, we can solve the $k$-MAX problem by using the standard CUCB algorithm. For space reasons, we provide the full algorithm and regret analysis in Appendix D and Appendix E.3. The estimates of parameters $\boldsymbol{p}$ and $\boldsymbol{v}$ are initialized to vectors with all elements equal to $1$. At each time step in which $v_j$ is observed to be the maximum value in the selected action, we update the estimate of $v_j$ and the estimates for $p_i$ for arms in the action set ordered before $j$. The algorithm maintains an upper confidence bound (UCB) for both parameters and feeds the UCB values to the approximation oracle to determine the next action.

## 3.3 ALGORITHM FOR UNKNOWN ORDERING OF VALUES

We now consider the case when the agent lacks knowledge about the ordering of values. This lack of knowledge significantly reduces the information that can be deduced from observed feedback. To understand this, let us examine the situation when the value $v_i$ of arm $a_i^v$ is yet observed. In the scenario where the ordering of values is known, the triggering of $a_i^z$ occurs when arm $i$ has higher value than that of the winning arm $j$. Consequently, we can deduce that $Z_i$ must be zero. However, in the case when the ordering is unknown, we cannot make this deduction. It remains uncertain whether $Z_i$ has a value of zero or one. Consider a round $t$ where action $S_t$ is played, and arm $j \in S_t$ emerges as the winner with value $v_j$. While the value-index pair $(v_j, j)$ is observed, the value $v_i$ for an arm $i \in S_t$ is still unknown. Consequently, we cannot determine whether $v_i < v_j$ or $v_i > v_j$. In the former case, the outcome of arm $i$ could be a non-zero value that remains unobserved, whereas it takes a value of zero in the latter case. Notably, the triggering of $a_i^z$ hinges on whether the value of $v_i$ is known. This aspect diverges from the CMAB-T framework. A conservative approach would be to apply CUCB algorithm for the known ordering case directly, and only increment the triggering count $T_i^v$ for arm $a_i^v$ after $v_i$ is observed. Parameters for arm $i \in \mathcal{A}$ would be updated only when $T_i^v \neq 0$. However, this conservative approach may cause the arms with low $p_i$ and low $v_i$ (bad arms with high risk and low reward) to be selected often, since their UCB values will be kept as $\bar{v}_i = 1$ and $\bar{p}_i = 1$. This would lead to significant regret.

We propose a variant of the CUCB algorithm outlined in Algorithm 3.1, which introduces the novel concept of *biased arm replacement*: We do not wait to update $p_i$ only after observing value $v_i$, but replace it with another arm with biased estimate of $\hat{v}_i = 1$. This approach treats $a_i^z$ as always triggered initially, allowing every arm a high chance of being a winner. As a result, the true winners gradually emerge, while arms whose values have not yet been observed are still given opportunities. Essentially, we extend CUCB by using a biased estimate of $(p_i', v_i')$ with $v_i' = 1$ to replace parameters $(p_i, v_i)$ of an unobserved arm. Our analysis shows that this bias can be controlled, and it still leads to a good regret bound. We present the result and an outline of the analysis below.

For each action $S \in \mathcal{F}$, we define the gap $\Delta_S = \max\{\alpha\text{OPT}(\boldsymbol{p}, \boldsymbol{v}) - r_S(\boldsymbol{p}, \boldsymbol{v}), 0\}$. We call an action $S$ *bad* if $\Delta_S > 0$. For each arm $i \in \mathcal{A}$ contained in at least one bad action, we define $\Delta_{\min}^i = \min\{\Delta_S \mid S : i \in S, q_i^{\boldsymbol{p}, S}, \tilde{q}_i^{\boldsymbol{p}, S} > 0, \Delta_S > 0\}$ and $\Delta_{\max}^i = \max\{\Delta_S \mid S : i \in$

$S, q_i^{\boldsymbol{p},S}, \tilde{q}_i^{\boldsymbol{p},S} > 0, \Delta_S > 0\}$. For every arm $i \in \mathcal{A}$ not contained in a bad action, we define $\Delta_{\min}^i = \infty$ and $\Delta_{\max}^i = 0$. Let $\Delta_{\min} = \min_{i \in [n]} \Delta_{\min}^i$ and $\Delta_{\max} = \max_{i \in [n]} \Delta_{\max}^i$.

**Theorem 3.4.** *For the $k$-MAX problem with max value-index feedback, under assumption that ordering of values is unknown to the algorithm a priori and $\Delta_{\min} > 0$, Algorithm 3.1 has the following distribution-dependent regret bound, for some constants $c_1$ and $c_2$,*

$$R(T) \leq c_1 \sum_{i=1}^{n} \left( \frac{k}{\Delta_{\min}^i} + \log \left( \frac{k}{\Delta_{\min}^i} + 1 \right) \right) \log(T) + c_2 \sum_{i=1}^{n} \left( \left( \log \left( \frac{k}{\Delta_{\min}^i} + 1 \right) + 1 \right) \Delta_{\max} + 1 \right).$$

Theorem 3.4 implies a regret upper bound of $O((nk/\Delta_{\min}) \log(T))$. This result aligns with the bound for the simpler case presented in Theorem D.1, up to constant factors. Furthermore, Theorem 3.4 implies a $\tilde{O}(\sqrt{nkT})$ distribution-independent regret bound (Appendix E.5). For modifications to Algorithm 3.1 aimed at improving the dependency of the regret upper bound on $k$, please refer to Appendix E.6.

**Proof sketch** We outline the proof of Theorem 3.4. The complete proof can be found in Appendix E.5. Our problem does not fit into the standard CMAB-T framework. As discussed earlier, the algorithm assumes that $a_i^z$ is always triggered and takes value zero. However, this may not be accurate when $v_i$ is actually less than the value of the winning arm. Consequently, the estimates produced by the algorithm are biased. We cannot simply apply the regret result of CMAB-T or follow its analysis to reach our desired outcome. To address this challenge, we introduce the concept of *biased arm replacement*. In each round $t$, for every arm $i$ with parameters $(p_i, v_i)$ and $T_{i,t}^v = 0$, we replace it with another arm having biased parameters $(p_i', v_i')$, where $p_i' = p_i v_i$ and $v_i' = 1$. We give subscript $t$ to $(\boldsymbol{p}', \boldsymbol{v}')$ in the following analysis as their values depend on $T_{i,t}^v$. The two arms have the same expected outcome but their effect on the reward among $k$ selected arms are different yet satisfying an order property, which is shown in the following lemma.

**Lemma 3.5.** *For every set $S \in \mathcal{F}$, $r_S(\boldsymbol{p}, \boldsymbol{v}) \leq r_S(\boldsymbol{p}', \boldsymbol{v}')$.*

Next, we consider the contribution of action $S_t$ to the regret, denoted as $\Delta_{S_t}$, under the assumption that the approximation oracle performs well ($\mathcal{F}_t$ : $r_{S_t}(\bar{\boldsymbol{p}}_t, \bar{\boldsymbol{v}}_t) \geq \alpha \, \mathrm{OPT}(\bar{\boldsymbol{p}}_t, \bar{\boldsymbol{v}}_t)$), where $\bar{\boldsymbol{p}}_t, \bar{\boldsymbol{v}}_t$ are the values of $\bar{\boldsymbol{p}}, \bar{\boldsymbol{v}}$ at the end of round $t$. By Lemma 3.5, for each $t$ such that $1 \leq t \leq T$, we have

$$\alpha \, \mathrm{OPT}(\boldsymbol{p}_t', \boldsymbol{v}_t') \geq \alpha \, \mathrm{OPT}(\boldsymbol{p}, \boldsymbol{v}). \tag{3.2}$$

Thus,

$$\begin{aligned}
\Delta_{S_t} &\leq \alpha \, \mathrm{OPT}(\boldsymbol{p}_t', \boldsymbol{v}_t') - r_{S_t}(\boldsymbol{p}, \boldsymbol{v}) \\
&\leq \alpha \, \mathrm{OPT}(\boldsymbol{p}_t', \boldsymbol{v}_t') - r_{S_t}(\boldsymbol{p}, \boldsymbol{v}) + r_{S_t}(\bar{\boldsymbol{p}}_t, \bar{\boldsymbol{v}}_t) - \alpha \, \mathrm{OPT}(\bar{\boldsymbol{p}}_t, \bar{\boldsymbol{v}}_t) \\
&\leq r_{S_t}(\bar{\boldsymbol{p}}_t, \bar{\boldsymbol{v}}_t) - r_{S_t}(\boldsymbol{p}, \boldsymbol{v}) = (r_{S_t}(\bar{\boldsymbol{p}}_t, \bar{\boldsymbol{v}}_t) - r_{S_t}(\boldsymbol{p}_t', \boldsymbol{v}_t')) + (r_{S_t}(\boldsymbol{p}_t', \boldsymbol{v}_t') - r_{S_t}(\boldsymbol{p}, \boldsymbol{v})).
\end{aligned}$$

The first inequality arises from condition (3.2), the second from the approximation oracle, and the third from the monotonicity of $r_S$ in $\boldsymbol{p}$ and $\boldsymbol{v}$. We refer to the term inside the first bracket as the regret caused by estimation error $\Delta_{S_t}^e$, and the term inside the second bracket as the regret caused by replacement error $\Delta_{S_t}^r$. To derive a tight regret upper bound, we show that the regret bound for the estimation error dominates that of the replacement error. This justifies the intuition behind using replacement arms. Now, let us examine these two terms separately.

By Lemma 3.3, we have

$$\Delta_{S_t}^e \leq \sum_{i \in S_t} q_i^{\boldsymbol{p}',S} \bar{v}_{i,t} (\bar{p}_{i,t} - p_{i,t}').$$

Note that we do not need to include the $v_i$ term, as $v_{i,t}' = \bar{v}_i = 1$ for all $i$ when $v_i$ is not observed, and $v_{i,t}' = \bar{v}_i = v_i$ after $v_i$ is observed. In both cases, there is no estimation error for $v_i$.

We apply Lemma 3.3 to obtain

$$\Delta_{S_t}^r \leq 2 \sum_{i \in S_t} q_i^{\boldsymbol{p},S} v_{i,t}' (p_i - p_{i,t}') + \sum_{i \in S_t} \tilde{q}_i^{\boldsymbol{p},S} (v_{i,t}' - v_i).$$

To sum up, we have

$$\Delta_{S_t} \leq \sum_{i \in S_t} q_i^{\boldsymbol{p}',S} \bar{v}_{i,t} (\bar{p}_{i,t} - p_{i,t}') + 2 \sum_{i \in S_t} q_i^{\boldsymbol{p},S} v_{i,t}' (p_i - p_{i,t}') + \sum_{i \in S_t} \tilde{q}_i^{\boldsymbol{p},S} (v_{i,t}' - v_i).$$

---

**Algorithm 4.1** CUCB algorithm for arbitrary distributions with finite supports

---

1: For $i \in \mathcal{A}$, $\sigma(i) \leftarrow 0$       ▷ Number of known values for arm $i$
2: For $i \in \mathcal{A}$, $T_{i,0}^z \leftarrow 0$, $T_{i,0}^v \leftarrow 0$       ▷ Number of triggering times for the fictitious arm
3: For $i \in \mathcal{A}$, $\hat{p}_{i,0} \leftarrow 1$, $\hat{v}_{i,0} \leftarrow 1$       ▷ Initial estimates of parameters for the fictitious arm
4: **for** $t = 1, 2, \ldots$ **do**
5:      For $i \in \mathcal{A}$ and $j \in [\sigma(i)]_0$, $\rho_{i,j} \leftarrow \sqrt{\frac{3 \log(t)}{2 T_{i,j}}}$, $\tilde{\rho}_{i,j} \leftarrow \mathbf{1}\{T_{i,j}^v = 0\}$ ▷ Confidence radius of parameters
6:      For $i \in \mathcal{A}$ and $j \in [\sigma(i)]_0$, $\bar{p}_{i,j} \leftarrow \min\{\hat{p}_{i,j} + \rho_{i,j}, 1\}$, $\bar{v}_{i,j} \leftarrow \min\{\hat{v}_{i,j} + \tilde{\rho}_{i,j}, 1\}$      ▷ UCB indices
7:      use Eq. (4.2) to convert the binary form of $\bar{\boldsymbol{p}}$ (treated as $\tilde{\boldsymbol{p}}$) to the multi-valued form $\check{\boldsymbol{p}}$ (treated as $\boldsymbol{p}$)
8:      $S \leftarrow \text{Oracle}(\check{\boldsymbol{p}}, \bar{\boldsymbol{v}})$       ▷ $k$-MAX PTAS oracle (Chen et al., 2016a)
9:      Play $S$ and observe winner index $i^*$ and value $v$
10:      **if** $v \notin \{\hat{v}_{i^*,j}, j \in [\sigma(i^*)]\}$ **then**
11:         $\sigma(i^*) \leftarrow \sigma(i^*) + 1$,    $T_{i^*,\sigma(i^*)}^z \leftarrow 0$,    $T_{i^*,\sigma(i^*)}^v \leftarrow 1$,    $\hat{v}_{i^*,\sigma(i^*)} \leftarrow v$
12:      **end if**
13:      For $i \in S$ and $j \in [\sigma(i)]_0$ such that $\hat{v}_{i,j} \geq v$: $T_{i,j}^z \leftarrow T_{i,j}^z + 1$
14:      For $i \in S$ and $j \in [\sigma(i)]_0$ such that $\hat{v}_{i,j} > v$: $\hat{p}_{i,j} \leftarrow (1 - 1/T_{i,j}^z)\hat{p}_{i,j}$
15:      For $i \in S$ and $j \in [\sigma(i)]_0$ such that $\hat{v}_{i,j} = v$: $\hat{p}_{i,j} \leftarrow (1 - 1/T_{i,j}^z)\hat{p}_i + 1/T_{i,j}^z$
16: **end for**

---

We can bound the first term by following the proof of the regret bound for the standard CMAB-T problem, stated in Theorem C.1 for completeness. To see this, recall that we have reset the count $T_i$ and the estimate $p_i$ at the time observing $v_i$. This is because $p'_{i,t} = p_i v_i$ when $v_i$ is unknown and $p'_{i,t} = p_i$ afterwards. However, for both stages, our estimates are accurate such that they always lie within the confidence interval which decreases as the counter number increases.

For the second term, we note that $p'_{i,t} = p_i v_i$ in the first stage, and $p'_{i,t} = p_i$ after observing $v_i$. Therefore, the contribution to regret by the second term is zero in the second stage. For the first stage, this term can be analyzed in a similar way as the last term. The key observation is that $p_i - p'_{i,t} = p_i(1 - v_i) \leq p_i$. This eliminates a large regret factor that would be introduced if we use the conservative CUCB approach as describe before.

Finally, we note that the analysis for the last summation term is the same as the case where the ordering of values is known, since there is no change to the triggering process of arms in $\mathcal{V}$.

Summing up the resulting bounds over time horizon $T$, we show that the first term dominates the other two terms, which leads to our theorem.

## 4    ARBITRARY DISTRIBUTIONS OF ARM OUTCOMES WITH FINITE SUPPORTS

We consider the more general case of arbitrary discrete distributions of arm outcomes with finite supports, which accommodates the case of binary distributions as a special case. We show that it is possible to represent a variable corresponding to an arm outcome with a set of binary variables. This allows us to extend to the general case of discrete distributions with finite supports.

To see this, let $X_i$ be a random variable with an arbitrary discrete distribution with finite support as defined in Section 2. Recall that $\Pr[X_i = v_{i,j}] = p_{i,j}$, for $j \in \{0, 1, \ldots, s_i\}$, where $v_{i,j} \in [0, 1]$ and $v_{i,0} = 0$. Let $X_{i,1}, \ldots, X_{i,s_i}$ be independent binary random variables such that $X_{i,j}$ takes value $v_{i,j}$ with probability $\tilde{p}_{i,j}$, and value $0$ otherwise. We can obtain the conversion between these two set of parameters by the following equations:

$$\tilde{p}_{i,j} = \frac{p_{i,j}}{1 - \sum_{l=j+1}^{s_i} p_{i,l}}, \forall j \in [s_i]. \quad (4.1) \quad p_{i,j} = \left( \prod_{l=j+1}^{s_i} (1 - \tilde{p}_{il}) \right) \tilde{p}_{ij}, \forall j \in [s_i]. \quad (4.2)$$

It can be readily checked that $\max\{X_{i,j} \mid j \in [s_i]\}$ has the same distribution as random variable $X_i$. In this way, we establish the equivalence between binary variables and any discrete variables with a finite support in terms of the max operator. This means that we can use our algorithm to solve the $k$-MAX problem for any discrete distributions of arm outcomes with finite supports. We present Algorithm 4.1 for the general discrete distributions with finite support. The algorithm is an extension of Algorithm 3.1, adapted to allow unknown support sizes of arm distributions.

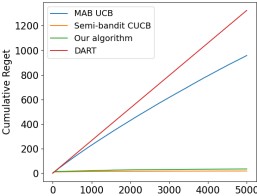 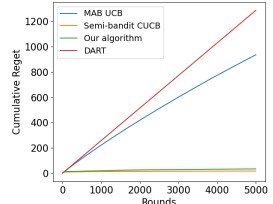 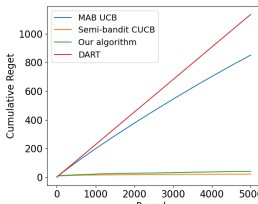

Figure 1: Regret for Algorithm 3.1 and benchmarks used for comparison, for different distributions of arm outcomes as defined in Appendix F.

Recall that we work with binary arms with outcomes according to $\{X_{i,j}, i \in [n], j \in [s_i]\}$ where $s_i + 1$ is the support size of $X_i$. To handle unknown support size $s_i$, we introduce a counter $\sigma(i)$ to denote the number of observed values of $X_i$ and dynamically maintain a list of these observed values. We increase this counter and reset the triggering times and probability estimates for the $\sigma(i)$-th arm whenever we observe a new value for $X_i$. On the other hand, we use a fictitious arm (denoted with index 0) with value 1 as a placeholder for those arms whose values remain unobserved. Since we do not know the support size, we always keep this fictitious arm and update its probability estimates just as a normal binary arm. If an observed value of $X_i$ is 1, we apply the deterministic tie-breaking rule to enforce that the fictitious arm is always ordered higher than the real binary arm with value 1 (see Appendix B on tie-breaking). In the algorithm, we use notation $[m]_0 = \{0, 1, 2, \ldots, m\}$.

Note that we convert UCBs of the binary arms to multi-valued forms according to relationship in Equation (4.2) and use the $k$-MAX PTAS in (Chen et al., 2016a) as the offline oracle. We cannot use the dynamic programming algorithm in (Du et al., 2024), because if we use the binary form, we need an oracle such that for each binary arm $(i, j)$ with outcome $X_{i,j}$, if $(i, j)$ is selected, then all $(i, j')$ for $j' \in [s_i]$ must also be selected because they correspond to the same base arm $X_i$. Due to the space constraint, we provide the theoretical regret guarantee in Appendix E.7.

## 5 NUMERICAL RESULTS

We perform experiments to evaluate the performance of Algorithm 3.1 on specific problem instances. We compare our algorithm with three baseline methods: the well-known UCB algorithm treating each set of size $k$ as one arm, the standard CUCB algorithm for semi-bandit CMAB (Chen et al., 2013), and DART (Agarwal et al., 2021b) for full-bandit CMAB. We use the greedy algorithm as the offline oracle. The code we use is available at: https://github.com/Sketch-EXP/kmax. We consider settings with $n = 9$ arms and sets of cardinality $k = 3$. We tested on three different arm distributions such that a single arm changes from one scenario to another. We expect the algorithm to perform well for all three cases. For space reasons, we show the detailed setup in Appendix F. We run each experiment for a horizon time of $T = 5000$. In each round, we select arms according to the offline oracle and sample their values for updates. We compare the reward to that of the optimal set $S^*$. We repeat the experiments 20 times and average the cumulative regrets.

**Results** Figure 1 presents the regrets of Algorithm 3.1 and three baseline methods across the three scenarios. It is evident that our algorithm achieves significantly lower regrets compared to the UCB algorithm and DART. Furthermore, our regret curve closely aligns with that of the CUCB method under the semi-bandit CMAB setting, indicating that we do not incur substantial losses despite receiving much less feedback.

## 6 CONCLUSION

We investigated the $k$-MAX combinatorial multi-armed bandit problem under maximum value-index feedback. This feedback, situated between semi-bandit and full-bandit feedback, provides less information compared to semi-bandit feedback. For the binary arm case, we propose a CUCB-based algorithm that introduces a novel concept of biased arm replacement, and for the general finite support case, we reduce it to the binary case and uses a fictitious arm to handle unknown support sizes. Through rigorous analysis, we demonstrated that our algorithms guarantee a regret upper bound comparable to that observed under semi-bandit feedback, albeit with some constant factors.

Future research could explore the $k$-MAX problem under full-bandit feedback, as well as extending the $k$-MAX objective to other variants such as the second max or soft max objectives. Furthermore, there is potential value in considering other CMAB problems under feedback scenarios between semi-bandit and full-bandit feedback, where biased arm replacement may also be applicable.

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

## SUPPLEMENTARY MATERIALS

## A    FURTHER DISCUSSION ON MOTIVATION

Our work focuses on $k$-MAX bandit for general arm distributions with finite, unknown support. We further discuss the real-world implications of our assumption regarding the arm support.

**Unknown support**    Firstly, we argue that in some cases, the agent has no knowledge of the feedback scale. For example, in bidding or donation scenarios, the agent may not have access to the internal values of the bidders or donors beforehand. Similarly, in sensor detection, the agent does not know the informativeness of sensors before receiving signals.

Moreover, we would like to emphasize the distinction between the choices that the platform offers and the internal values of the items that a user actually perceives. On most systems, choices may be offered on a 1-10 or 1-100 scale. If we treat this as the case of known support size, the support size would be 10 or 100, respectively. However, the user or customer may value a given item with only one or a few possible values. By treating these possible values as unknown supports, we may have a much smaller support size to deal with on each random variable, resulting in smaller regret.

**Finite support**    We also argue that our assumption of finite support is applicable across various applications, as categorical ratings or ranks are commonly used in the real world. Additionally, we note the possibility of extending our results to accommodate continuous reward distributions by discretizing continuous arm distributions. However, discretization may introduce a tradeoff between discretization granularity and regret bound.

Furthermore, it is worth noting that the observation made in the previous paragraph applies here as well. In many cases, even though the support size may be infinite based on the values offered by the platform (e.g., real-valued feedback), the actual value of the customer for an item may only have a few possible internal values. Therefore, treating the variable as having finite and unknown support size could effectively reduce regret in numerous practical scenarios.

## B    FURTHER DISCUSSION ON TIE-BREAKING

In Section 3, we assume that all values $v_1, v_2, \ldots, v_n$ are unique, and state that this is equivalent to allowing for non-unique values with a deterministic tie-breaking rule. In this appendix, we provide a more formal argument for this equivalence.

Deterministic tie-breaking means that whenever two arms $i$ and $j$ appear in the same action $S$ and their outcomes have the same values $v_i = v_j$, and this outcome is the maximum among all outcomes of arms in $S$, we deterministically decide which of $i$ or $j$ would win according to a predefined order. Without loss of generality, we assume the predefined order dictates that the smaller indexed arm always wins in tie-breaking; that is, if $v_i = v_j$, then arm $i$ wins if $i < j$.

For any problem instance with the above tie-breaking rule, we can convert it to another instance where all values are unique, showing that the two instances essentially behave the same. The conversion involves adding lower-order digits to the current value of $v_i$'s to make all values unique while respecting the deterministic tie-breaking rule. The actual conversion is as follows: Suppose a problem instance in the deterministic tie-breaking setting is $(\boldsymbol{p}, \boldsymbol{v})$, and without loss of generality, assume that $v_1 \geq v_2 \geq \cdots \geq v_n$. Let $\varepsilon_0 > 0$ be an arbitrarily small value. Let $\varepsilon = \varepsilon_0/n$. Then we construct a new problem instance $(\boldsymbol{p}', \boldsymbol{v}')$, such that $\boldsymbol{p}' = \boldsymbol{p}$, and for all $i$, $v_i' = v_i + (n - i)\varepsilon$. With this construction, we know that if $v_i > v_j$, then $i < j$, and $v_i' = v_i + (n - i)\varepsilon > v_j + (n - j)\varepsilon = v_j'$; if $v_i = v_j$ and $i < j$, then $v_i' = v_i + (n - i)\varepsilon > v_j + (n - j)\varepsilon = v_j'$. This means we have all unique values with $v_1' > v_2' > \cdots > v_n'$, and the value order respects the previous value order together with the deterministic tie-breaking rule. Because $\boldsymbol{p}' = \boldsymbol{p}$, we know that for any action $S$, the winning index distribution for these two instances must be the same. Moreover, for every $i$, $|v_i' - v_i| = (n - i)\varepsilon < \varepsilon_0$, which means the reward value of the new instance is arbitrarily close to the original instance. Therefore, through the above construction, we show that any problem instance under the deterministic tie-breaking rule can be viewed as an instance with unique values that are arbitrarily close to the original values. This demonstrates that the problem instances under the deterministic tie-breaking rule can be treated equivalently as instances with unique values with essentially the same result.

In the recommender system setting, deterministic tie-breaking models the scenario where customers instinctively seek additional clues or information when faced with two similar items to make a selection, rather than relying solely on chance. There may be a discrepancy between the value feedback based on the choices offered by the agent and the internal values of the items, depending on the granularity of the feedback scale. For example, consider four items A, B, C, and D presented to a user on a five-star scale. Initially, A, B, and C are shown, and the user favors A, rating it 4 stars. Later, when B, C, and D are shown, the user prefers B and also rates it 4 stars. Although the user gives the same value feedback to both A and B, there may be additional factors not captured in the feedback, leading to deterministic tie-breaking when choosing between A and B simultaneously. Random tie-breaking, on the other hand, may be of interest primarily for mathematical curiosity or theoretical exploration.

## C CMAB-T FRAMEWORK AND ADDITIONAL NOTATION

We review the framework and results for the classical CMAB problem with triggered arms considered by (Wang & Chen, 2017). In this problem, the expected reward is a function of action $S$ and expected values of arm outcomes $\boldsymbol{\mu}$. We denote the probability that action $S$ triggers arm $i$ as $p_i^{\boldsymbol{\mu},S}$. It is assumed that in each round, the values of triggered arms are observed by the agent. The CUCB algorithm (Chen et al., 2013) is used to estimate the expectation vector $\boldsymbol{\mu}$ directly from samples.

The following theorem pertains to the standard CMAB problem with triggered arms (CMAB-T). It is assumed that the CMAB-T problem instance satisfies monotonicity and 1-norm TPM bounded smoothness (Definition 3.2 with $b_i = B$ for all $i \in [n]$). We will use some of the proof steps and the result of this theorem in proofs of our results.

**Theorem C.1** (Theorem 1 in (Wang & Chen, 2017)). *For the CUCB algorithm on a CMAB-T problem instance satisfying monotonicity and 1-norm TPM bounded smoothness, we have the following distribution-dependent regret bound,*

$$R(T) \leq 576 B^2 k \left( \sum_{i=1}^{n} \frac{1}{\Delta_{\min}^i} \right) \log(T) + \left( \sum_{i=1}^{n} \left( \log \left( \frac{2Bk}{\Delta_{\min}^i} + 1 \right) + 2 \right) \right) \frac{\pi^2}{6} \Delta_{\max} + 4Bn$$

*where* $\Delta_{\min} = \inf_{S:i \in S, p_i^{\boldsymbol{\mu},S} > 0, \Delta_S > 0} \Delta_S$.

We next give various definitions used in our analysis. The definitions are given specifically for binary distributions of arm outcomes.

We define two sets of *triggering probability (TP) groups*. Let $j$ be a positive integer. For the set $\mathcal{Z}$ of arms we define the triggering probability group $\mathcal{S}_{i,j}^z$ as

$$\mathcal{S}_{i,j}^z = \{ S \in \mathcal{F} \mid 2^{-j} < q_i^{\boldsymbol{p},S} \leq 2^{-j+1} \}.$$

We define the triggering probability group $\mathcal{S}_{i,j}^v$ for the set $\mathcal{V}$ of arms similarly. We note that the triggering probability groups divide actions that trigger arm $i$ into separated groups such that the actions in the same group contribute similarly to the regret bound.

Let $N_{i,j,t}^z$ be the counter of the cumulative number of times $i$ in TP group $\mathcal{S}_{i,j}^z$ is selected at the end of round $t$. Similarly, we use $N_{i,j,t}^v$ to denote the counter of the cumulative number of times $i$ in TP group $\mathcal{S}_{i,j}^v$ is selected at the end of round $t$.

We define the *event-filtered regret* for a sequence of events $\{\mathcal{E}_t\}$ as

$$R(T, \{\mathcal{E}_t\}) = T \alpha \operatorname{OPT}(\boldsymbol{p}, \boldsymbol{v}) - \mathbb{E} \left[ \sum_{t=1}^{T} \mathbf{1}(\mathcal{E}_t) r_{S_t}(\boldsymbol{p}, \boldsymbol{v}) \right]$$

which means that the regret is accounted for in round $t$ only if event $\mathcal{E}_t$ occurs in round $t$.

We next define four good events as follows:

**E1** The approximation oracle works well, i.e.

$$\mathcal{F}_t = \{ r_{S_t}(\bar{\boldsymbol{p}}, \bar{\boldsymbol{v}}) \geq \alpha \operatorname{OPT}(\bar{\boldsymbol{p}}, \bar{\boldsymbol{v}}) \}.$$

---

**Algorithm D.1** CUCB algorithm for known ordering of values

1: For $i \in \mathcal{A}$, $T_i \leftarrow 0$           ▷ Number of triggering times for $a_i^z$
2: For $i \in \mathcal{A}$, $\hat{p}_i \leftarrow 1$, $\hat{v}_i \leftarrow 1$        ▷ Initial estimation of parameters
3: **for** $t = 1, 2, \ldots, T$ **do**
4:   For $i \in \mathcal{A}$, $\rho_i \leftarrow \sqrt{\frac{3 \log(t)}{2 T_i}}$        ▷ Confidence radius values
5:   For $i \in \mathcal{A}$, $\bar{p}_i \leftarrow \min\{\hat{p}_i + \rho_i, 1\}$, $\bar{v}_i \leftarrow \hat{v}_i$     ▷ UCB values
6:   $S \leftarrow \text{Oracle}(\bar{\boldsymbol{p}}, \bar{\boldsymbol{v}})$      ▷ Oracle decides the next action
7:   Play $S$ and observe winner index $i^*$ and value $v_{i^*}$
8:   Update $\hat{v}_{i^*}$ for winner arm $i^*$: $\hat{v}_{i^*} \leftarrow v_{i^*}$
9:   For $i \in S$ such that $i \leq i^*$: $T_i \leftarrow T_i + 1$
10:   For $i \in S$ such that $i < i^*$: $\hat{p}_i \leftarrow (1 - 1/T_i)\hat{p}_i$
11:   $\hat{p}_{i^*} \leftarrow (1 - 1/T_{i^*})\hat{p}_{i^*} + 1/T_{i^*}$
12: **end for**

---

**E2** The parameter vector $\boldsymbol{p}$ is estimated well, i.e. for every $i \in [n]$ and $t \geq 1$,

$$\mathcal{N}_{i,t} = \{|\hat{p}_{i,t-1} - p_i| < \rho_{i,t}\}$$

where $\hat{p}_{i,t-1}$ is the estimator of $p_i$ at round $t$ and $\rho_{i,t} := \sqrt{3 \log(t)/(2T_{i,t-1})}$. We denote this event with $\mathcal{N}_t$.

**E3** Triggering is nice for $\mathcal{Z}$ given a set of integers $\{j_{\max}^i\}_{i \in [n]}$, i.e. for every TP group $\mathcal{S}_{i,j}^z$ defined by arm $i$ and $1 \leq j \leq j_{\max}^i$, under the condition $\sqrt{6 \log(t)/(N_{i,j,t-1}^z/3)2^{-j}} \leq 1$, it holds $T_{i,t-1}^z \geq \frac{1}{3} N_{i,j,t-1}^z 2^{-j}$. We denote this event with $\mathcal{G}_t^z$.

**E4** Triggering is nice for $\mathcal{V}$, i.e. for every arm $i \in \mathcal{A}$, under the condition $N_{i,j,t-1}^z \geq 3 p_i^{-1} \log(t) 2^j$, there is $T_{i,t-1}^v \neq 0$. Equivalently, we can define this event in terms of TP group $S_{i,j}^v$ by removing the factor of $p_i^{-1}$, i.e., under the condition $N_{i,j,t-1}^v \geq 3 \log(t) 2^j$. We denote this event with $\mathcal{G}_t^v$.

Note that E1, E2 and E3 are also used in (Wang & Chen, 2017), and event E4 is newly defined for our problem. We can easily show the following bound by using the Hoeffding's inequality.

**Lemma C.2.** *For each round $t \geq 1$, it holds $\Pr(\neg \mathcal{G}_t^v) \leq n/t^2$.*

## D   CUCB ALGORITHM FOR KNOWN ORDERING OF VALUES

We propose the CUCB algorithm defined in Algorithm D.1. As discussed in the main text, the estimates of parameters $\boldsymbol{p}$ and $\boldsymbol{v}$ are initialized to vectors with all elements equal to $1$. At each time step in which $v_j$ is observed to be the maximum value in the selected action, we update the estimate of $v_j$ and the estimates for $p_i$, for arms in the action set ordered before $j$. The algorithm maintains an upper confidence bound (UCB) for both parameters and feeds the UCB values to the approximation oracle to determine the next action.

**Theorem D.1.** *For the $k$-MAX problem with max value-index feedback, under assumption that ordering of values is a priori known to the algorithm and $\Delta_{\min} > 0$, Algorithm D.1 has the following distribution-dependent regret bound,*

$$R(T) \leq c_1 \sum_{i=1}^{n} \left( \frac{k}{\Delta_{\min}^i} + \log\left( \frac{k}{\Delta_{\min}^i} + 1 \right) \right) \log(T) + c_2 \sum_{i=1}^{n} \left( \left( \log\left( \frac{k}{\Delta_{\min}^i} + 1 \right) + 1 \right) \Delta_{\max} + 1 \right),$$

*where $c_1$ and $c_2$ are some positive constants.*

In Theorem D.1, the only term in the regret bound that depends on horizon time $T$ is the first summation term. In this summation term, the summands have two terms, one scaling linearly with $k/\Delta_{\min}^i$ and other scaling logarithmically with $k/\Delta_{\min}^i$, which are due to uncertainty of parameters $\boldsymbol{p}$ and $\boldsymbol{v}$, respectively. Hence, we may argue that the uncertainty about values of parameters $\boldsymbol{p}$ has more effect on regret than uncertainty about values of parameters $\boldsymbol{v}$. The regret bound in Theorem D.1 implies a $\tilde{O}(\sqrt{T})$ distribution-independent regret bound.

To see how the regret analysis of the algorithm can be decomposed to a CMAB-T problem, we consider the contribution of each action to regret, i.e, $\Delta_{S_t} = \max\{\alpha \, \mathrm{OPT}(\boldsymbol{p}, \boldsymbol{v}) - r_{S_t}(\boldsymbol{p}, \boldsymbol{v}), 0\}$. Let $\mathcal{F}_t$ be the good event $\{r_{S_t}(\bar{\boldsymbol{p}}, \bar{\boldsymbol{v}}) \geq \alpha \, \mathrm{OPT}(\boldsymbol{p}, \boldsymbol{v})\}$ meaning that the approximation oracle works well. By the smoothness condition, under $\mathcal{F}_t$ we have

$$\Delta_{S_t} \leq r_{S_t}(\bar{\boldsymbol{p}}_t, \bar{\boldsymbol{v}}_t) - r_{S_t}(\boldsymbol{p}, \boldsymbol{v}) \leq \sum_{i \in S_t} q_i^{\boldsymbol{p}, S} \bar{v}_{i,t}(\bar{p}_{i,t} - p_i) + \sum_{i \in S_t} \tilde{q}_i^{\boldsymbol{p}, S}(\bar{v}_{i,t} - v_i).$$

Clearly, the first term corresponds to regret from the set of arms $\mathcal{Z}$, and the second term corresponds to regret from the set of arms $\mathcal{V}$. We bound $\Delta_{S_t}$ by bounding the two summation terms individually. The first summation term is standard in existing literature. For bounding the second term, we need to make extra steps as our estimates for $v_i$ are not more and more accurate as the number of selections of arm $i$ increases. The UCB value for $v_i$ remains at the upper bound value 1 until arm $a_i^v$ is triggered once and we then know the exact value of $v_i$. We show the proof of Theorem D.1 in Appendix E.3.

## E  PROOFS

### E.1  PROOF OF LEMMA 3.1

Without loss of generality, consider $S = \{1, \ldots, k\}$ under assumption $v_1 \geq \cdots \geq v_k$. The expected regret can be written as

$$r_S(\boldsymbol{p}, \boldsymbol{v}) = p_1 v_1 + (1 - p_1) p_2 v_2 + \cdots + (1 - p_1) \cdots (1 - p_{k-1}) p_k v_k.$$

It is clear from the expression that $r_S(\boldsymbol{p}, \boldsymbol{v})$ is monotonic increasing in $v_i$.

Let us consider $r_S(\boldsymbol{p}, \boldsymbol{v})$ for arbitrarily fixed $\boldsymbol{v}$. Taking derivative with respect to $p_i$, we have

$$\frac{d}{dp_i} r_S(\boldsymbol{p}, \boldsymbol{v}) = (1 - p_1) \cdots (1 - p_{i-1}) \left[ v_i - p_{i+1} v_{i+1} - \left( \sum_{j > i} (1 - p_{i+1}) \cdots (1 - p_j) p_{j+1} v_{j+1} \right) \right].$$

We claim that the term inside the bracket is non negative. This can be shown as follows,

$$
\begin{aligned}
& v_i - p_{i+1} v_{i+1} - \left( \sum_{j > i} (1 - p_{i+1}) \cdots (1 - p_j) p_{j+1} v_{j+1} \right) \\
\geq\ & v_i(1 - p_{i+1}) - (1 - p_{i+1}) p_{i+2} v_{i+2} - (1 - p_{i+1})(1 - p_{i+2}) p_{i+3} v_{i+3} - \cdots \\
\geq\ & (1 - p_{i+1})(1 - p_{i+2}) \cdots (1 - p_{k-1})(v_i - p_k v_k) \\
\geq\ & (1 - p_{i+1})(1 - p_{i+2}) \cdots (1 - p_k) v_i \\
\geq\ & 0.
\end{aligned}
$$

Thus, the reward function is monotonic increasing in $p_i$.

### E.2  PROOF OF LEMMA 3.3

For the purpose of this proof only, we assume that the arms are ordered in decreasing order of values. This applies to both $\boldsymbol{v}$ and $\boldsymbol{v}'$. Recall that

$$r_S(\boldsymbol{p}, \boldsymbol{v}) = \sum_{i \in S} v_i p_i \prod_{j \in S : j < i} (1 - p_j).$$

Let $\boldsymbol{p} = (p_1, \ldots, p_k)$ and $\boldsymbol{p}' = (p'_1, \ldots, p'_k)$, and $\boldsymbol{p}^{(0)} = \boldsymbol{p}'$, $\boldsymbol{p}^{(j)} = (p_1, \ldots, p_j, p'_{j+1}, \ldots, p'_k)$, for $1 \leq j < k$, and $\boldsymbol{p}^{(k)} = \boldsymbol{p}$. Similarly, let $\boldsymbol{v} = (v_1, \ldots, v_k)$ and $\boldsymbol{v}' = (v'_1, \ldots, v'_k)$, and $\boldsymbol{v}^{(0)} = \boldsymbol{v}'$, $\boldsymbol{v}^{(j)} = (v_1, \ldots, v_j, v'_{j+1}, \ldots, v'_k)$, for $1 \leq j < k$, and $\boldsymbol{v}^{(k)} = \boldsymbol{v}$.

Let us define

$$f(\boldsymbol{p}^{(j)}, \boldsymbol{v}^{(j)}) = p_1 v_1 + \cdots + (1 - p_1) \cdots (1 - p_{j-1}) p_j v_j + (1 - p_1) \cdots (1 - p_j) p'_{j+1} v'_{j+1} + \cdots$$

Clearly, we have $r_S(\boldsymbol{p}, \boldsymbol{v}) = f(\boldsymbol{p}^{(k)}, \boldsymbol{v}^{(k)})$ and $r_S(\boldsymbol{p}', \boldsymbol{v}') = f(\boldsymbol{p}^{(0)}, \boldsymbol{v}^{(0)})$. Note that

$$f(\boldsymbol{p}^{(j-1)}, \boldsymbol{v}^{(j-1)}) = p_1 v_1 + \cdots + (1 - p_1) \cdots (1 - p_{j-1}) p'_j v'_j + (1 - p_1) \cdots (1 - p'_j) p'_{j+1} v'_{j+1} + \cdots$$

The only difference is caused by position $j$. By definition of triggering probabilities $q_i^{\boldsymbol{p},S}$ and $\tilde{q}_i^{\boldsymbol{p},S}$ we can write,

$$
\begin{aligned}
|f(\boldsymbol{p}^{(j)}, \boldsymbol{v}^{(j)}) - f(\boldsymbol{p}^{(j-1)}, \boldsymbol{v}^{(j-1)})| =& |q_j^{\boldsymbol{p},S}(p_j v_j - p_j' v_j' - \sum_{i>j}(1 - p_{j+1}') \ldots (1 - p_{i-1}') p_i' v_i' (p_j - p_j')| \\
\leq& q_j^{\boldsymbol{p},S} p_j |v_j - v_j'| + q_j^{\boldsymbol{p},S} v_j' |p_j - p_j'| \\
&+ q_j^{\boldsymbol{p},S}(p_{j+1}' v_{j+1}' + (1 - p_{j+1}') p_{j+2}' v_{j+2}' + \ldots)|p_j - p_j'| \\
\leq& 2 q_j^{\boldsymbol{p},S} v_j' |p_j - p_j'| + \tilde{q}_j^{\boldsymbol{p},S} |v_j - v_j'|
\end{aligned}
$$

where the first inequality is due to the triangle inequality and the second inequality is due to the monotonicity property. If $\boldsymbol{p} \geq \boldsymbol{p}'$, we only need to consider the first two terms in the second line.

Now we note that

$$
\begin{aligned}
|r_S(\boldsymbol{p}, \boldsymbol{v}) - r_S(\boldsymbol{p}', \boldsymbol{v}')| =& |f(\boldsymbol{p}^{(k)}, \boldsymbol{v}^{(k)}) - f(\boldsymbol{p}^{(0)}, \boldsymbol{v}^{(0)})| \\
\leq& \sum_{j=1}^{k} |f(\boldsymbol{p}^{(j)}, \boldsymbol{v}^{(j)}) - f(\boldsymbol{p}^{(j-1)}, \boldsymbol{v}^{(j-1)})|
\end{aligned}
$$

Summing up over $j$ we obtain the statement of the lemma.

### E.3 Proof of Theorem D.1

We consider the contribution of action $S_t$ to regret $\Delta_{S_t}$, for every $t \geq 1$. Let $M_i := \Delta_{\min}^i$. Recall that $\Delta_S = \max\{\alpha \cdot \text{OPT}_{\boldsymbol{p},\boldsymbol{v}} - r_S(\boldsymbol{p}, \boldsymbol{v}), 0\}$. Assume that $\Delta_{S_t} \geq M_{S_t}$ where $M_S = \max_{i \in S} M_i$. Note that if $\Delta_{S_t} < M_{S_t}$ then $\Delta_{S_t} = 0$, since we have either an empty set, or $\Delta_{S_t} < M_{S_t} < M_i$ for some $i \in S_t$.

By the smoothness condition, we have

$$
\Delta_{S_t} \leq r_{S_t}(\bar{\boldsymbol{p}}_t, \bar{\boldsymbol{v}}_t) - r_{S_t}(\boldsymbol{p}, \boldsymbol{v}) \leq \sum_{i \in S_t} q_i^{\boldsymbol{p},S} \bar{v}_{i,t}(\bar{p}_{i,t} - p_i) + \sum_{i \in S_t} \tilde{q}_i^{\boldsymbol{p},S}(\bar{v}_{i,t} - v_i).
$$

Since $\Delta_{S_t} \geq M_{S_t}$, we add and subtract $M_{S_t}$ from the last expression and we have,

$$
\begin{aligned}
\Delta_{S_t} \leq& -M_{S_t} + 2 \left( \sum_{i \in S_t} q_i^{\boldsymbol{p},S} \bar{v}_{i,t}(\bar{p}_{i,t} - p_i) + \sum_{i \in S_t} \tilde{q}_i^{\boldsymbol{p},S}(\bar{v}_{i,t} - v_i) \right) \\
\leq& 2 \left( \sum_{i \in S_t} q_i^{\boldsymbol{p},S} \bar{v}_{i,t}(\bar{p}_{i,t} - p_i) - \frac{M_i}{4k} \right) + 2 \left( \sum_{i \in S_t} \tilde{q}_i^{\boldsymbol{p},S}(\bar{v}_{i,t} - v_i) - \frac{M_i}{4k} \right).
\end{aligned}
$$

Let us call the first term $\Delta_{S_t}^p$ and the second term $\Delta_{S_t}^v$. We bound $\Delta_{S_t}$ by bounding the two summation terms individually.

Note that we can bound the term $\Delta_{S_t}^p$ following the same procedure as in the proof for Theorem C.1. However, we cannot use the same procedure for $\Delta_{S_t}^v$. The key difference is that our estimate for $v_i$ will not be more and more accurate as the number of selections of arm $i$ increases. We know the exact value of $v_i$ as soon as it is triggered once. We assume that arm $i$ is in TP group $S_{i,j}^v$. Let $j_i$ be the index of the TP group with $S_t \in S_{i,j_i}^v$. We take $j_{\max}^i = \log(4k/M_i + 1)$.

- Case 1: $1 \leq j_i \leq j_{\max}^i$. In this case, $\tilde{q}_i^{\boldsymbol{p},S} \leq 2 \cdot 2^{-j_i}$,

$$
\tilde{q}_i^{\boldsymbol{p},S}(\bar{v}_{i,t} - v_i) \leq 2 \cdot 2^{-j_i} \mathbb{1}\{T_{i,t}^v = 0\}
$$

Under the good event $\mathcal{G}_t^v$, when $N_{i,j_i,t-1}^v \geq 3 \log(t) \cdot 2^j$, the contribution to regret is zero. Otherwise, it is bounded by,

$$
\tilde{q}_i^{\boldsymbol{p},S}(\bar{v}_{i,t} - v_i) \leq 2 \cdot 2^{-j_i}.
$$

- Case 2: $j_i \geq j_{\max}^i + 1$. In this case,

$$
\tilde{q}_i^{\boldsymbol{p},S}(\bar{v}_{i,t} - v_i) \leq 2 \cdot 2^{-j_i} \leq \frac{M_i}{4k}.
$$

Thus the term does not contribute to regret in this case.

We next calculate the filtered regret under the good events mentioned above and the event that $\Delta_{S_t} \geq M_{S_t}$. Note that

$$R(T, \{\Delta_{S_t} \geq M_{S_t}\}, \mathcal{F}_t, \mathcal{N}_t, \mathcal{G}_t^z, \mathcal{G}_t^v) \leq \sum_{t=1}^{T} \Delta_{S_t}^p + \sum_{t=1}^{T} \Delta_{S_t}^v.$$

For the event E3 we set $j_{\max}^i = \log(4k/M_i + 1)$. By Theorem C.1 in Appendix C, we know that under good events E1, E2 and E3, the first term is bounded by

$$\sum_{t=1}^{T} \Delta_{S_t}^p \leq 1152k \left( \sum_{i=1}^{k} \frac{1}{M_i} \right) \log(T) + 4n$$

and, otherwise, the corresponding filtered regrets are

$$R(T, \neg \mathcal{F}_t) \leq (1 - \beta) T \cdot \Delta_{max}$$

$$R(T, \neg \mathcal{N}_t) \leq \frac{\pi^2}{3} n \Delta_{\max}$$

$$R(T, \neg \mathcal{G}_t^z) \leq \frac{\pi^2}{6} \left( \sum_{i=1}^{n} \log \left( \frac{4k}{M_i} + 1 \right) \right) \Delta_{\max}.$$

Now, we focus on bounding the contribution of the second term to regret. Note that under event E4,

$$\sum_{t=1}^{T} \Delta_{S_t}^v = \sum_{i=1}^{n} \sum_{j=1}^{\infty} \sum_{s=0}^{N_{i,j,T-1}^v} \kappa_{j_i, T}(M_i, s)$$

where

$$\kappa_{j,T}(M, s) = \begin{cases} 2 \cdot 2^{-j} & \text{if } s < 3 \log(T) \cdot 2^j \\ 0 & \text{if } s \geq 3 \log(T) \cdot 2^j \end{cases}.$$

For every arm $i$ and $j \geq 1$, we have

$$\sum_{s=0}^{N_{i,j,T-1}^v} \kappa_{j_i, T}(M_i, s) \leq \sum_{s=0}^{3 \log(T) \cdot 2^{j_i}} \kappa_{j_i, T}(M_i, s) = 6 \log(T).$$

Hence, the contribution of the second term to regret is bounded by

$$\sum_{i=1}^{n} \sum_{j=1}^{\infty} \sum_{s=0}^{N_{i,j,T-1}^v} \kappa_{j_i, T}(M_i, s) \leq 6 \left( \sum_{i=1}^{n} \log \left( \frac{4k}{M_i} + 1 \right) \right) \log(T).$$

The filtered regrets for the case when event E4 fails to hold is bounded by,

$$R(T, \neg \mathcal{G}_t^v) \leq \sum_{i=1}^{T} \Pr(\neg \mathcal{G}_t^v) \Delta_{max} \leq \frac{\pi^2}{6} n \Delta_{\max}.$$

We obtain the distribution-dependent regret bound by adding up the filtered regrets calculated above. The corresponding distribution-independent regret bound is implied by taking $M_i = \sqrt{16nk/T}$ for every $i \in [n]$.

### E.4 PROOF OF LEMMA 3.5

Without loss of generality assume that $S = [k]$ and $v_1 \geq v_2 \geq \cdots \geq v_k$. Recall that we can write

$$r_S(\boldsymbol{p}, \boldsymbol{v}) = p_1 v_1 + (1 - p_1) p_2 v_2 + \ldots + (1 - p_1) \ldots (1 - p_{k-1}) p_k v_k.$$

Now, for $\boldsymbol{p} = (p_1, \ldots, p_k)$ and $\boldsymbol{p}' = (p'_1, \ldots, p'_k)$, let

$$\boldsymbol{p}^{(j)} = (p'_1, \ldots, p'_j, p_{j+1}, \ldots, p_k)$$

and define similarly $\boldsymbol{v}^{(j)}$ for $\boldsymbol{v}$ and $\boldsymbol{v}'$. After changing $p_1$ to $p'_1 = p_1 v_1$ and $v_1$ to $v'_1 = 1$,

$$r_S(\boldsymbol{p}^{(1)}, \boldsymbol{v}^{(1)}) = p_1 v_1 + (1 - p_1 v_1) p_2 v_2 + \cdots + (1 - p_1 v_1) \cdots (1 - p_{k-1}) p_k v_k.$$

Clearly we have $r_S(\boldsymbol{p}^{(1)}, \boldsymbol{v}^{(1)}) \geq r_S(\boldsymbol{p}, \boldsymbol{v})$. Following the same argument, we can see that $r_S(\boldsymbol{p}^{(2)}, \boldsymbol{v}^{(2)}) \geq r_S(\boldsymbol{p}^{(1)}, \boldsymbol{v}^{(1)})$. Continuing this way to $r_S(\boldsymbol{p}^{(k)}, \boldsymbol{v}^{(k)})$ we can prove the lemma.

### E.5 PROOF OF THEOREM 3.4

By the RTPM smoothness condition, as discussed in the main text, we have

$$\Delta_{S_t} \le \sum_{i \in S_t} q_i^{\boldsymbol{p}',S} \bar{v}_{i,t}(\bar{p}_{i,t} - p'_{i,t}) + 2 \sum_{i \in S_t} q_i^{\boldsymbol{p},S} v'_{i,t}(p_i - p'_i) + \sum_{i \in S_t} \tilde{q}_i^{\boldsymbol{p},S}(v'_{i,t} - v_i).$$

**Key step: bounding the contribution of each action to regret** Let $M_i = \Delta_{\min}^i$. Assume that $\Delta_{S_t} \ge M_{S_t}$ where $M_S = \max_{i \in S} M_i$. As in the known value ordering case, we can bound $\Delta_{S_t}$ such that,

$$\Delta_{S_t} \le - M_{S_t} + 2 \left( \sum_{i \in S_t} q_i^{\boldsymbol{p}',S} \bar{v}_{i,t}(\bar{p}_{i,t} - p'_{i,t}) + 2 \sum_{i \in S_t} q_i^{\boldsymbol{p},S} v'_{i,t}(p_i - p'_i) + \sum_{i \in S_t} \tilde{q}_i^{\boldsymbol{p},S}(v'_{i,t} - v_i) \right)$$

$$\le 2 \left( \sum_{i \in S_t} q_i^{\boldsymbol{p}',S} \bar{v}_{i,t}(\bar{p}_{i,t} - p'_{i,t}) - \frac{M_i}{8k} \right) + 4 \left( \sum_{i \in S_t} q_i^{\boldsymbol{p},S} v'_{i,t}(p_i - p'_i) - \frac{M_i}{8k} \right)$$

$$+ 2 \left( \sum_{i \in S_t} \tilde{q}_i^{\boldsymbol{p},S}(v'_{i,t} - v_i) - \frac{M_i}{8k} \right). \tag{E.1}$$

Let $j_i$ be the index of the TP group of $S_t$ such that $S_t \in S_{i,j_i}^z$. We bound $\Delta_{S_t}$ by bounding the three summation terms in (E.1) separately.

**Bounding the first term.** Recall that we reset $T_{i,t}^z$ and $N_{i,j,t}^z$ at the time we observe $v_i$. This is because $p'_{i,t} = p_i v_i$ and $v'_{i,t} = 1$ when $v_i$ is unknown (first stage) and $p'_{i,t} = p_i$ and $v'_{i,t} = v_i$ after observing $v_i$ (second stage). A key observation is that within both stages our estimates are accurate in the sense that under event E2, the approximation error decreases as the counter number increases in the following way.

$$\bar{p}_{i,t} - p'_{i,t} \le 2\rho_i = 2\sqrt{\frac{3 \log(t)}{2 T_{i,t-1}^z}}.$$

We note that for the second stage where $v_i$ is observed, this term is similar to the $\Delta_{S_t}^p$ term in the known value ordering case. Specifically, under event E3,

$$\bar{p}_{i,t} - p'_{i,t} \le \min \left\{ \sqrt{\frac{18 \log(t)}{2^{-j_i} \cdot N_{i,j_i,t-1}^z}}, 1 \right\}$$

thus

$$q_i^{\boldsymbol{p}',S} \bar{v}_{i,t}(\bar{p}_{i,t} - p'_{i,t}) \le \min \left\{ \sqrt{\frac{72 \cdot 2^{-j_i} \log(T)}{N_{i,j_i,t-1}^z}}, 2 \cdot 2^{-j_i} \right\}.$$

For the event E3, let $j_{\max}^i = \log(8k/M_i + 1)$. In the case $j_i \ge j_{\max}^i + 1$, this term does not contribute to regret as we have,

$$q_i^{\boldsymbol{p}',S}(\bar{p}_{i,t} - p'_{i,t}) \le 2 \cdot 2^{-j_i} \le \frac{M_i}{8k}$$

Similarly, for $1 \le j_i \le j_{\max}^i$, there is no contribution to regret if $N_{i,j,t-1}^z \ge l_{j_i,T}(M_i)$ where

$$l_{j,T}(M) = \left\lfloor \frac{4608 \cdot 2^{-j} k^2 \log(T)}{M^2} \right\rfloor.$$

For the first stage, we note that the event E3 is not required to hold, as we treat $a_i^z$ as always triggered and the triggering is always nice for $\mathcal{Z}$ in the first stage. Thus for the first stage we have,

$$q_i^{\boldsymbol{p}',S} \bar{v}_{i,t}(\bar{p}_{i,t} - p'_{i,t}) \le \min \left\{ \sqrt{\frac{6 \log(T)}{T_{i,t-1}^z}}, 1 \right\}$$

This term does not contribute to regret if $T_{i,t-1}^z \ge l'_T(M_i)$ where

$$l'_T(M) = \left\lfloor \frac{384 k^2 \log(T)}{M^2} \right\rfloor.$$

Bounding the second term. Take $j_{\max}^i = \log(8k/M_i + 1)$. Similarly as the first term, in the case $j_i \geq j_{\max}^i + 1$, the contribution to regret is non-positive. For $1 \leq j_i \leq j_{\max}^i$, as $p'_{i,t} = p_i v_i$, we have

$$q_i^{\boldsymbol{p},S} v'_{i,t}(p_i - p'_{i,t}) \leq 2 \cdot 2^{-j_i} p_i (1 - v_i).$$

Under the event E4, we know that the contribution to regret is zero if $N_{i,j_i,t-1}^z \geq 3p_i^{-1} \log(T) \cdot 2^j$. Otherwise, it is upper bounded by $2 \cdot 2^{-j_i} p_i$.

Bounding the third term. Take $j_{\max}^i = \log(8k/M_i + 1)$. In the case $j_i \geq j_{\max}^i + 1$, the contribution to regret is non-positive. For $1 \leq j_i \leq j_{\max}^i$, we have $\tilde{q}_i^{\boldsymbol{p},S} \leq 2 \cdot 2^{-j_i} p_i$, thus

$$\tilde{q}_i^{\boldsymbol{p},S}(\bar{v}_{i,t} - v_i) \leq 2 \cdot 2^{-j_i} p_i \tilde{\rho}_i = 2 \cdot 2^{-j_i} p_i \cdot \mathbb{1}\{T_{i,t}^v = 0\}.$$

Under the event E4, we know that the contribution to regret is zero if $N_{i,j_i,t-1}^z \geq 3p_i^{-1} \log(T) \cdot 2^j$. Otherwise, it is upper bounded by

$$\tilde{q}_i^{\boldsymbol{p},S}(\bar{v}_{i,t} - v_i) \leq 2 \cdot 2^{-j_i} p_i$$

Note that this bound is the same as the bound for the second term.

**Summing over the time horizon** Next, we sum up $\Delta_{S_t}$ over time $T$ and calculate the filtered regret under the above mentioned good events and the event that $\Delta_{S_t} \geq M_{S_t}$, i.e, $R(\{\Delta_{S_t} \geq M_{S_t}\}, \mathcal{F}_t, \mathcal{N}_t, \mathcal{G}_t^z, \mathcal{G}_t^v)$.

By equation (E.1), we know that the filtered regret can be upper bounded by sum of three terms over the time horizon $T$. By Theorem C.1 in Appendix C, we know under good events E1, E2 and E3, the second stage of the first term is bounded by

$$2\left(\sum_{i \in S_t} q_i^{\boldsymbol{p},S} \bar{v}_{i,t}(\bar{p}_{i,t} - p'_{i,t}) - \frac{M_i}{8k}\right) \leq 2304k\left(\sum_{i=1}^n \frac{1}{M_i}\right)\log(T) + 4n$$

and, otherwise, the corresponding filtered regrets are bounded by,

$$R(T, \neg\mathcal{F}_t) \leq (1-\beta)T\Delta_{\max}$$

$$R(T, \neg\mathcal{N}_t) \leq \frac{\pi^2}{3}n\Delta_{\max}$$

$$R(T, \neg\mathcal{G}_t^z) \leq \frac{\pi^2}{6}\sum_{i=1}^n \log\left(\frac{8k}{M_i} + 1\right)\Delta_{\max}.$$

To bound the first term, we also need to derive a bound for the first stage when the value is not observed. This is bounded by,

$$2\left(\sum_{i \in S_t} q_i^{\boldsymbol{p}',S}(\bar{p}_{i,t} - p'_{i,t}) - \frac{M_i}{8k}\right) \leq \sum_{i=1}^n \sum_{s=1}^{l'_T(M_i)} 2\sqrt{\frac{6\log(T)}{s}} = 64k\left(\sum_{i=1}^n \frac{1}{M_i}\right)\log(T).$$

Following similar analysis as for the $\Delta_{S_t}^v$ term of Theorem D.1, we have that

$$2\sum_{t=1}^T \left(\sum_{i \in S_t} \tilde{q}_i^{\boldsymbol{p},S}(v'_{i,t} - v_i) - \frac{M_i}{8k}\right) \leq 12\sum_{i=1}^n \log\left(\frac{8k}{M_i} + 1\right)\log(T).$$

Similarly, we can bound

$$4\sum_{t=1}^T \left(\sum_{i \in S_t} q_i^{\boldsymbol{p},S} v'_{i,t}(p_i - p'_{i,t}) - \frac{M_i}{8k}\right) \leq 24\sum_{i=1}^n \log\left(\frac{8k}{M_i} + 1\right)\log(T).$$

The filtered regret for the case where event E4 fails to hold is bounded by,

$$R(T, \neg\mathcal{G}_t^v) \leq \sum_{i=1}^T \Pr(\neg\mathcal{G}_t^v)\Delta_{\max} \leq \frac{\pi^2}{6}n\Delta_{\max}.$$

We obtain the distribution-dependent regret by adding up the filtered regrets calculated above. Similarly as before, the distribution-independent regret is implied by taking $M_i = \sqrt{64nk/T}$ for every $i \in [n]$. The distribution-independent regret bound is $\tilde{O}(\sqrt{T})$, the same as cascading bandits upper bound.

$$R(T) \leq c_1 \left( \sqrt{nkT} + \log\left( \sqrt{nkT} + 1 \right) \right) \log T + c_2 \left( \left( \log\left( \sqrt{nkT} + 1 \right) + 1 \right) \Delta_{\max} + 1 \right).$$

### E.6 FURTHER IMPROVEMENT OF REGRET BOUND DEPENDENCY ON $k$

We can further improve the regret upper bound dependency on $k$ from $k$ to $\log(k)$ using TPVM condition in (Liu et al., 2022). We outline the steps and changes as follows.

**Algorithm** In the algorithm, besides maintaining a confidence bound for the mean value of $p$, we also maintain a confidence bound for its variance as Algorithm 1 in (Liu et al., 2022).

**Verification for TPVM** Following the notations in (Liu et al., 2022), let $p' = p + \zeta + \eta$, then

$$|r_S(\boldsymbol{p}, \boldsymbol{v}) - r_S(\boldsymbol{p}', \boldsymbol{v}')| \leq B_v \sqrt{\sum_{i \in S} q_i^{\boldsymbol{p},S} \frac{\zeta_i^2}{(1 - p_i)p_i}} + B_1 \sum_{i \in S} q_i^{\boldsymbol{p},S} \eta_i + \sum_{i \in S} \tilde{q}_i^{\boldsymbol{p},S} |v_i - v_i'|$$

which satisfies TPVM with respect to the base arm set $\mathcal{Z}$ with $(B_v, B_1, \lambda) = (2, 0, 1)$.

To see this, consider the first term of the RTPM condition in Lemma 3.3,

$$\begin{aligned}
2 \sum_{i \in S} q_i^{\boldsymbol{p},S} v_i' |p_i - p_i'| &= 2 \sum_{i \in S} \sqrt{q_i^{\boldsymbol{p},S} p_i (1 - p_i)} \sqrt{\frac{v_i'^2 q_i^{\boldsymbol{p},S}}{p_i(1 - p_i)} |p_i - p_i'|^2} \\
&\leq 2 \sum_{i \in S} \sqrt{\tilde{q}_i^{\boldsymbol{p},S}} \sqrt{\frac{v_i'^2 q_i^{\boldsymbol{p},S}}{p_i(1 - p_i)} |p_i - p_i'|^2} \leq 2 \sqrt{\sum_{i \in S} q_i^{\boldsymbol{p},S} \frac{|p_i - p_i'|^2}{(1 - p_i)p_i}}
\end{aligned}$$

where the first inequality is due to definition of triggering probabilities, and the second inequality is due to the fact that $\sum_{i \in S} \tilde{q}_i^{\boldsymbol{p},S} \leq 1$ and $v_i' \leq 1$.

**Regret analysis** Recall that we have $\Delta_{S_t} \leq \Delta_{S_t}^e + \Delta_{S_t}^r$. We make changes to bound the estimation error $\Delta_{S_t}^e$. We note that the replacement error will not be affected. We define $\mathcal{F}_e = \{\Delta_{S_t} \leq 2\Delta_{S_t}^e\}$ and $\mathcal{F}_r = \{\Delta_{S_t} \leq 2\Delta_{S_t}^r\}$. We note that $R(T) \leq R(T, \mathcal{F}_e) + R(T, \mathcal{F}_r)$. We consider the filtered regret under $\mathcal{F}_e$.

We replace the confidence radius specified in the good event (E2) by the confidence radius specified in Lemma 8 of (Liu et al., 2022),

(E2) The parameter vector $\boldsymbol{p}$ is estimated well, i.e. for every $i \in [n]$ and $t \geq 1$,

$$\mathcal{N}_{i,t} = \{|\hat{p}_{i,t-1} - p_i| < \rho_{i,t}\}$$

where $\hat{p}_{i,t-1}$ is the estimator of $p_i$ at round $t$ and $\rho_{i,t} := 4\sqrt{3}\sqrt{\frac{p_i(1-p_i)\log(t)}{T_{i-1,t}}} + \frac{28\log(t)}{T_{t-1,t}}$.

Under $\mathcal{F}_e$ and (E2), we have

$$\Delta_{S_t} \leq 8\sqrt{3}\sqrt{\sum_{i \in S_t} \min\left\{\frac{\log(t)}{T_{i-1,t}}, 1/28\right\} q_i^{\boldsymbol{p},s}} + 56 \sum_{i \in S_t} \min\left\{\frac{\log(t)}{T_{i-1,t}}, 1/28\right\} q_i^{\boldsymbol{p},s}$$

We note that we can still use the technique of arm replacement to show that within both stages our estimates for $p$ are accurate. As explained in the proof of Theorem 3.4, the first stage corresponds to a degenerated case where arms are always triggered, and the second stage corresponds to the case of standard probabilistically triggered arms. Following steps of Lemma 10 and Lemma 11 in Appendix C.3 in (Liu et al., 2022), we can conclude that

$$R(T, \mathcal{F}_e) \leq O\left( \sum_{i=1}^{n} \log\left( \frac{k}{\Delta_{\min}^i} \right) \frac{\log k}{\Delta_{\min}^i} \log(T) \right).$$

And from Theorem 3.4, we know that

$$R(T, \mathcal{F}_e) \leq O\left(\sum_{i=1}^{n} \log\left(\frac{k}{\Delta_{\min}^i}\right) \log(T)\right).$$

Therefore, we improve the regret upper bound from $O((k/\Delta)\log T)$ to $O((\log k/\Delta)\log T)$.

### E.7 THEORETICAL GUARANTEE OF ALGORITHM 4.1

We claim the comparable regret bound upper bound as in Theorem 3.4 holds for Algorithm 4.1.

**Theorem E.1.** *For the $k$-MAX problem with max value-index feedback, under the general case of arbitrary discrete distributions of arm outcomes with finite support as defined in Section 2 and under assumption $\Delta_{\min} > 0$, Algorithm 4.1 has the following $(1 - \varepsilon, 1)$ distribution-dependent regret bound when using the k-MAX PETAS oracle from (Chen et al., 2016a) that guarantees $(1 - \varepsilon)$ approximation,*

$$R(T) \leq c_1 \sum_{i=1}^{n} \left(\frac{k}{\Delta_{\min}^i} + \log\left(\frac{k}{\Delta_{\min}^i} + 1\right)\right) s_i \log(T) + c_2 \sum_{i=1}^{n} s_i \left(\left(\log\left(\frac{k}{\Delta_{\min}^i} + 1\right) + 1\right) \Delta_{\max} + 1\right).$$

We explain the derivation steps in details.

Let $0 = v_{i,0} < v_{i,1} < \cdots < v_{i,s_i}$ denote values of the support of distribution of $X_i$, where $s_i$ is a positive integer, and $s_i + 1$ is the support size. Recall that $p_{i,j} = \Pr[X_i = v_{i,j}]$ for $j \in \{0, 1, \ldots, s_i\}$, with $0 < \sum_{j=1}^{s_i} p_{i,j} \leq 1$.

By Equation (4.1), for each variable $X_i$, we can replace it with a set of binary variables $X_{i,j}$'s. We provide more detailed explanation to justify this reduction. For ease of discussion, we restate the equation here:

$$\tilde{p}_{i,j} = \begin{cases} \frac{p_{i,j}}{1 - \sum_{l=j+1}^{s_i} p_{i,l}} & \text{if } 1 \leq j < s_i \\ p_{i,s_i} & \text{if } j = s_i. \end{cases}$$

The key is to verify that $Y_i = \max\{X_{i,j} \mid j \in [s_i]\}$ has the same distribution as the original random variable $X_i$. In fact, for every $v_{i,j}$ in $X_i$'s support, the probability of $Y_i = v_{i,j}$ is equal to the probability that for all $j' > j$, $X_{i,j'} = 0$ and $X_{i,j} = v_{i,j}$. By Equation (E.1), the latter probability is $\prod_{j' > j}(1 - \tilde{p}_{i,j'}) \cdot \tilde{p}_{i,j} = \prod_{j'=j+1}^{s_i}(1 - \frac{p_{i,j'}}{1 - \sum_{l=j'+1}^{s_i} p_{i,l}}) \cdot \frac{p_{i,j}}{1 - \sum_{l=j+1}^{s_i} p_{i,l}} = \prod_{j'=j+1}^{s_i}(\frac{1 - \sum_{l=j'}^{s_i} p_{i,l}}{1 - \sum_{l=j'+1}^{s_i} p_{i,l}}) \cdot \frac{p_{i,j}}{1 - \sum_{l=j+1}^{s_i} p_{i,l}} = p_{i,j}$. Therefore, the probability of $Y_i = v_{i,j}$ is the same as the probability of $X_i = v_{i,j}$. Since $Y_i$ can only take values from the support values of $X_i$, we can see that $Y_i$ and $X_i$ have the same distribution.

Given the above fact, we can see that for each variable $X_i$, we can replace it with variables $X_{i,j}$'s, so that when we select a set $S$ of $X_i$'s, $\max\{X_i, i \in S\}$ has the same distribution as $\max\{X_{i,j}, i \in S, j \in [s_i]\}$, because $\max\{X_i, i \in S\} = \max\{Y_i, i \in S\} = \max\{\max\{X_{i,j} \mid j \in [s_i]\}, i \in S\} = \max\{X_{i,j}, i \in S, j \in [s_i]\}$.

Consider the contribution of action $S_t$ to regret. By the above reduction,

$$\Delta_{S_t} \leq \alpha \, \text{OPT}(\boldsymbol{p}, \boldsymbol{v}) - r_{S_t}(\boldsymbol{p}, \boldsymbol{v})$$
$$= \alpha \, \text{OPT}(\tilde{\boldsymbol{p}}, \boldsymbol{v}) - r_{S_t}(\tilde{\boldsymbol{p}}, \boldsymbol{v}).$$

For the remaining steps, we will treat the regret $r_{S_t}(\boldsymbol{p}, \boldsymbol{v})$ as that of the set of equivalent binary arms.

We note that we convert the UCBs of the binary arms $(\bar{\boldsymbol{p}}, \bar{\boldsymbol{v}})$ to their multi-valued forms and use the $k$-MAX PTAS offline oracle. Again, if the approximation oracle works well, by Equation (E.1), we have $r_{S_t}(\bar{\boldsymbol{p}}, \bar{\boldsymbol{v}}) \geq \alpha \, \text{OPT}(\bar{\boldsymbol{p}}, \bar{\boldsymbol{v}})$. Therefore, by the arm replacement implementing on the set of binary arms,

$$\Delta_{S_t} \leq (r_{S_t}(\bar{\boldsymbol{p}}, \bar{\boldsymbol{v}}) - r_{S_t}(\boldsymbol{p}_t', \boldsymbol{v}_t')) + (r_{S_t}(\boldsymbol{p}_t', \boldsymbol{v}_t') - r_{S_t}(\boldsymbol{p}, \boldsymbol{v}))$$

By the RTPM condition applied to the set of binary arms,

$$\Delta_{S_t} \leq \sum_{i \in S_t} \sum_{j=0}^{s_i} q_{i,j}^{\boldsymbol{p}',S} \bar{v}_{i,j,t}(\bar{p}_{i,j,t} - p_{i,j,t}') + 2 \sum_{i \in S_t} \sum_{j=0}^{s_i} q_{i,j}^{\boldsymbol{p},S} v_{i,j,t}'(p_{i,j} - p_{i,j,t}') + \sum_{i \in S_t} \sum_{j=0}^{s_i} \tilde{q}_{i,j}^{\boldsymbol{p},S}(v_{i,j,t}' - v_{i,j}).$$

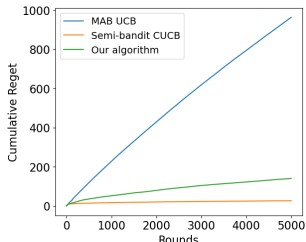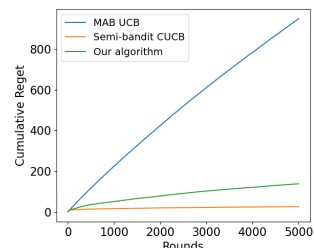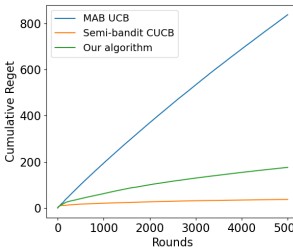

Figure 2: Regret for Algorithm 4.1 and considered benchmark algorithms, for different distributions of arm outcomes.

In Algorithm 4.1, we relax the assumption of knowing the support size of arm distributions by introducing a counter $\sigma(i)$ for each arm $X_i$. This means for some $i$ and binary arms $\{X_{i,j}, j = 1, \ldots, s_i - \sigma(i)\}$ whose values $v_{i,j}$ have not been observed, we replace them with a single fictitious binary arm, instead of $s_i - \sigma(i)$ fictitious arms. This is clearly justified, as the replaced arms have values $v'_{i,j} = 1$. For each term inside the summations, we can bound following the arguments as in Theorem 3.4. We can obtain the regret upper bound as claimed.

## F  SUPPLEMENTARY INFORMATION FOR SIMULATIONS

In this section, we provide supplementary information for Section 5 on numerical results.

**Setup**   We consider settings with $n = 9$ arms and sets of cardinality $k = 3$ for the following distributions of arms:

**D1**  $\boldsymbol{v} = (0.1, 0.2, \ldots, 0.9)^\top$. $\boldsymbol{p}$ is such that $p_i = 0.3$ for $1 \leq i \leq 6$, and $p_i = 0.5$ otherwise.

**D2**  Similar to D1, but we modify arm 1 to have a larger probability $p_1 = 0.9$.

**D3**  Similar to D1, but we modify the last arm to have a smaller probability $p_9 = 0.2$. Despite having the same expected value as arm 6, arm 9 is included in the optimal set.

Note that the optimal super arm is $S^* = \{7, 8, 9\}$ in all cases. Distributions D1, D2 and D3 represent different scenarios. D1 is the base case. In D2, there is a stable arm with low value, while in D3 there is a high-risk high-reward arm. Both are not easy to observe and cause challenges for our algorithm design, especially the latter type of arms, which can outperform less-risky arms under the maximum value reward function.

We also verify Algorithm 4.1 for the general case of arbitrary discrete distributions of arm outcomes with finite supports. For each arm distribution, we slightly vary the non-zero arm values by adding a small value perturbations to $v_i$, such that each arm $i$ takes two different non-zero values centered around $v_i$. We assume the small value perturbation follows Gaussian distribution with mean $v_i$ and standard deviation 0.03.

**Regret plots**   We show the regrets of Algorithm 3.1 and three baseline methods in Figure 1. The regret plots for the categorical settings using Algorithm 4.1 are shown in Figure 2. We plot the 1-approximation regrets instead of $(1 - 1/e)$-approximation regret as the offline greedy oracle performs much better than $(1 - 1/e)$-approximation in this case. We can see that our algorithm performs well, achieving much lower regrets in all cases.

**Discussion on DART**   We note that DART does not perform well in our binary setting. This can be attributed to mainly two reasons. As mentioned in the section of related works, DART assumes that arms can be ordered according to their mean rewards obtained by playing actions containing them. However, in our case, arm values cannot be determined in this way. DART is therefore inherently unsuitable for our tasks. Moreover, DART requires a threshold $\Delta$ to accept and reject. In our case, it is hard for DART to distinguish between good and bad arms. To reach a sufficiently small threshold, DART takes over 10000 rounds of iterations. On the other hand, our algorithm can correctly order arms at 5000 rounds, as shown in the regret plots. For these reasons, we do not include DART as a benchmark for the categorical setting.

