# OpenReview forum: "Combinatorial Bandits for Maximum Value Reward Function under Value-Index Feedback"
_ICLR.cc/2024/Conference — ICLR 2024 poster_

### Official Review · Reviewer_oHUd · 2023-10-27

**Soundness:** 3 good
**Presentation:** 3 good
**Contribution:** 2 fair
**Rating:** 6
**Confidence:** 3

**Summary:**

The paper focuses on solving a combinatorial multi-armed bandit problem that incorporates both maximum value and index feedback, a structure that falls between semi-bandit and full-bandit feedback scenarios. The authors introduce an algorithm and establish regret bounds for the case where arm outcomes are subject to arbitrary distributions with finite supports. They examine a broader range of arms and employ a smoothness condition to analyze regret. The algorithm achieves regret bounds that depend on the specific distribution as well as bounds that are distribution-independent, and these results are similar to those found in more informative semi-bandit feedback situations. Experimental evidence validates the algorithm's effectiveness.

**Strengths:**

The paper presents an innovative feedback structure called "max value-index feedback," positioning it between the well-explored semi-bandit and full-bandit feedback frameworks. This introduces a fresh approach to understanding combinatorial bandit problems. The authors establish regret bounds for their algorithm, encompassing situations that rely on distribution-specific and distribution-agnostic characteristics. These bounds are demonstrated to be on par with those obtained in more informative semi-bandit feedback settings. In the paper, algorithms are introduced for both binary and arbitrary discrete distributions of arm outcomes, incorporating the max value-index feedback, and they exhibit effective performance.

**Weaknesses:**

The known ordering case is a simple extension of the cascading bandit.This should be more emphasis that this is not a contribution but rather a simple first case for better understanding of the new and non-trivial unknown ordering case.

Although the paper presents the application scenarios that motivate it, it is necessary to explore in more detail how the proposed approach could be applied to a wider range of problems. The paper focuses on one specific problem, the k-MAX bandit problem with maximum value index return, and I'm not convinced that this is a fundamental step towards dealing with the full-bandit CMAB problem.

The paper is specifically centered on addressing a particular combinatorial multi-armed bandit problem that incorporates maximum value and index feedback. However, it does not delve into exploring how this approach might be applied to different types of bandit problems or alternative feedback structures. The paper does not thoroughly clarify the algorithm's limitations when dealing with diverse problem settings.

The paper assumes that arm outcomes follow arbitrary distributions with finite supports, allowing for a broad treatment of the problem. Nevertheless, it may not accurately capture the characteristics of real-world scenarios. The implications of these assumptions on the algorithm's performance and regret bounds are not exhaustively examined.

While the paper does provide experimental results that showcase the effectiveness of the proposed algorithm, these experiments are conducted on a limited scale and lack extensive benchmarking against other algorithms. To gain a more comprehensive understanding of the algorithm's performance, a more thorough evaluation on a wider range of problem instances and a comparison with existing approaches would be beneficial.

**Questions:**

What is the main obstacle in finding the lower bound for the exact problem setting you are considering ?

Would it be possible to extend this work to unbounded rewards ? Like Gaussian ones ?

---

> ### Author Response · Authors · 2023-11-20
> **Thanks for the valuable feedback!**
>
> We would like to thank the reviewer for positively appreciating the technical soundness and presentation of our paper. We give detailed comments on the points of concern and questions raised in the review. We update the appendix with more discussions and will incorporate all addressed points in the final version of our paper.
>
> P1 *Although the paper presents the application scenarios that motivate it, it is necessary to explore in more detail how the proposed approach could be applied to a wider range of problems.*
> P2 *The paper is specifically centered on addressing a particular combinatorial multi-armed bandit problem that incorporates maximum value and index feedback..*
>
> Yes, we acknowledge that the current work focuses on one specific problem. We will downplay a bit of this point of contribution, but we note that 1) the max value-index feedback assumes far less information than most of other types of feedback structures 2) the idea of arm equivalence and using fictitious arms can be possibly applied to solve CMAB problems of other non-linear rewards.
>
> P3 *The paper assumes that arm outcomes follow arbitrary distributions with finite supports, allowing for a broad treatment of the problem. Nevertheless, it may not accurately capture the characteristics of real-world scenarios.*
>
> We argue that our setting assuming finite support is not limited in the applications, as categorical ratings or ranks are commonly used in the real world. We note the possibility of extending our result to allow for continuous reward distributions by discretizing continuous arm distributions. However, discretization may incur a tradeoff between discretization granularity and regret bound. We add a section in the appendix giving more detailed comments on the real world implications of our assumptions on the arm support.
>
> P4 *While the paper does provide experimental results that showcase the effectiveness of the proposed algorithm, these experiments are conducted on a limited scale and lack extensive benchmarking against other algorithms.*
>
> The main goal of the numerical part is to show that our algorithm achieves good regret bound, as proved in the theoretical results. In particular, it is designed purposely to show that the algorithm works for scenarios with high-risk high-reward items or stable-valued items with low reward. We chose the benchmark algorithm carefully such that this assumption works. We will test for alternative algorithms such as DART where this assumption does not hold and add another plot for items of categorical distributions.
>
> P5 *What is the main obstacle in finding the lower bound for the exact problem setting you are considering?*
>
> We acknowledge that finding the lower bound for the exact problem setting remains an open problem. We note that related works such as Kveton et al 2015 and Wang and Chen 2017 use arguments similar to ours. The key is to find a special case of $k$-MAX problem such that items are almost always triggered but the probability parameters are never accurately estimated.
>
> P6 *Would it be possible to extend this work to unbounded rewards? Like Gaussian ones?*
>
> We believe that it is possible to extend the work to unbounded rewards like Gaussian. Unboundness is not a serious issue as Gaussian distribution is highly centred. There are prior works that extend from bounded rewards to unbounded rewards using concentration inequalities. We also argue that the infinite support size is not of a big concern. In many cases, even though the support size may be infinite from the values offered by the platform (e.g. real-valued feedback), the actual value of the customer on an item may only have a few possible internal values, and thus treating the variable as having finite and unknown support size could work well and save regret for many practical scenarios.

---

> > ### Comment · Reviewer_oHUd · 2023-11-22
> >
> > I acknowledge having read the rebuttal. I would like to thank the authors for their response to my questions.

---

### Official Review · Reviewer_AcYK · 2023-10-31

**Soundness:** 3 good
**Presentation:** 2 fair
**Contribution:** 3 good
**Rating:** 6
**Confidence:** 3

**Summary:**

This paper considers combinatorial bandits with maximum value and index feedback. The new feedback model lies somewhere in between semi-bandit and full-bandit feedback models. The new feedback model has applications in recommender systems. Assuming that the maximum value is unique, if we know the maximum value and the index of the base arm that achieves the maximum value, then we also know that all base arms selected in that particular round have reward realizations less than or equal to the maximum reward. The authors analyze the gap-dependent and gap-free regrets under base arms with finite support. They show that under these assumptions, the dependence of the regret on the batch size k and time matches with that of semi-bandit feedback algorithms.

**Strengths:**

This paper considers an important CMAB problem with real-world applications. There exists a wide spectrum of feedback models that lie between semi-bandit and full-bandit feedback models. Max-value feedback with an index is one of them that is mainly encountered in recommender systems. Although it seems to provide just a little more information than the full-bandit feedback setting, i.e., the index of the arm that achieves the max reward, correct utilization of this additional information results in performance almost matching with the semi-bandit feedback model.

The paper nicely develops the theory for the new setting by starting from the Bernoulli reward case with known support values. It starts building the algorithms by borrowing tools from the CMAB-T framework. In particular, extension to arbitrary distributions with finite supports by using the representation of the outcome of a single arm with finite support as multiple binary variables and using an appropriate action selection oracle is interesting. This equivalence allows the use of the techniques formed for Bernoulli rewards in addressing the more general and challenging case.

**Weaknesses:**

The most practically relevant setting for the proposed work seems to be the case with arbitrary distributions of arm outcomes with finite supports. Technical development in the paper starts from the simpler cases and gives the reader an expectation that the special cases are solved to provide intuition for the general case. Algorithm 4.1 looks like the pinnacle of the paper. However, is there any result (theoretical or experimental) related to this algorithm? If the regret bound in Theorem 3.4 still holds, putting proof of this would be nice. Moreover, simulations are concerned with the Bernoulli case. Verifying the performance of Algorithm 4.1 in simulations would be nice.

Another weakness is related to the deterministic tie-breaking rule, which is proposed as a way to accommodate non-unique arm reward values. I wonder if there is an application in which this tie-breaking is in the control of the decision-maker instead of its environment. More about this is asked in the questions part of the review.

Other weaknesses are mostly related to presentation issues and pinpointing the difference between the estimators used compared to the ones in the semi-bandit feedback CMAB-T setting. Please see the questions section.

**Questions:**

Based on my initial reading, there are several unclear parts in the paper. I will be able to provide a better assessment of the quality of the paper after the authors' feedback. Answers to the following questions are crucial for my reassessment.

- Section 3 rests on the assumption that for any action $S_t$, there is a unique arm achieving the maximum value among all arms in $S_t$. Then, it is argued that this can be extended to the non-unique values using a deterministic tie-breaking rule. My question is about implementation issues regarding this deterministic tie-breaking rule in a recommender system. A user who likes two of the shown movies out of k can randomly decide to watch one of them. Imposing a deterministic tie-breaking rule puts a restriction on the behavior of the user. I wonder if this tie-breaking rule can be justified in practice, specifically for applications in recommender systems, for instance, in the one that I mentioned above.

- How is $q^{\mu,S}_i$ related to $q^{p,S}_i$ and $\tilde{q}^{p,S}_i$?

- As a starting point, the authors extend the CUCB algorithm to accommodate for max value and index feedback for the case with Bernoulli rewards and known ordering of values. The claim is that most of the parts of the regret analysis of CUCB for CMAB-T remain valid under the new setting. Is it the case that in max-value max-index feedback, the estimates of base arm outcomes $p_i$ are still unbiased? This seems crucial for event E2 from the previous work Wang & Chen 2017 to apply, which achieves unbiasedness thanks to semi-bandit feedback.

- Please comment on the batch-size dependence of the distribution-free regret bound derived from B.1. Is it the same as cascading bandits upper bound?

- What about batch size independent regret bounds as in Liu et al. 2022, NeurIPS? Is it possible to obtain them under your current assumptions? Does triggering probability and variance modulated smoothness (TPVM) condition hold for your reward function? If so, what is the most general case in your paper so that this assumption holds? If this condition holds, then providing TPVM-based regret bounds will be good.

---

> ### Author Response · Authors · 2023-11-20
> **Thanks for the insightful feedback!**
>
> We thank the reviewer for the insightful feedback. We have addressed each point of concern and questions asked in the following. More updates are included in the appendix and we will incorporate all addressed points in the final version of our paper.
>
> P1 *Algorithm 4.1 looks like the pinnacle of the paper. However...*
>
> Yes, a comparable regret upper bound as in Theorem 3.4 holds for Algorithm 4.1: For the $k$-MAX problem with max value-index feedback, under general case of arbitrary discrete distributions of arm outcomes with finite	support as defined in Section 2 and under assumption $\Delta_{\min}>0$, Algorithm 4.1 has the following distribution-dependent regret bound,
> $$R(T)\leq c_1 \sum_{i=1}^n\left(\frac{k}{\Delta_{\min}^i} + \log\left(\frac{k}{\Delta_{\min}^i}+1\right)\right) s_i\log(T) +c_2 \sum_{i=1}^n s_i\left(\left(\log \left(\frac{ k}{\Delta_{\min}^i}+1\right)+1\right)\Delta_{\max}+1\right),$$
> We give further comments of the claim in the appendix.
>
> The current experimental setup is designed purposefully to show that the algorithm works for scenarios with high-risk high-reward items or stable-valued items with low reward. We will add another plot for items of categorical distributions to demonstrate that Algorithm 4.1 is an effective approach.
>
> P2 *I wonder if there is an application in which this tie-breaking..*
> P3 *My question is about implementation issues regarding this deterministic tie-breaking rule..*
>
> Thanks for asking this good question. In the recommender system setting, deterministic tie-breaking models the case that the customer is always intrinsically looking for other clues and information when facing two similar items to make a selection in his/her head, instead of purely relying on a coin toss in his/her head. There could be a difference between the value feedback based on the choices that the platform offers and the internal values of the items that a user actually perceives, which rely on the granularity of the feedback scale. Think of four items A, B, C, and D given to the user on a five-star scale. The first time, A, B, and C are shown and the user favors A and rates it 4 stars. The second time, B, C, and D are shown and the user favors B and rates it also 4 stars. The user gives both A and B the same value feedback, but there could be some confounding factors that are not captured in the value feedback, which implies a deterministic tie-breaking when facing A and B at the same time. Considering random tie-breaking is perhaps only purely for mathematical curiosity or interest.
>
> P4 *How is $q_i^{\mu,S}$ related to $q_i^{p,S}$ and $\tilde{q}_i^{p,S}$?*
>
> $q_i^{\mu,S}$ in Definition 3.2 is a general definition for the triggering probability of arm $i$ under action $S$ and a set of base arms $\mathcal{B}$ with expectation values $\mu$, and the reward function depends on $\mu$ only. $q_i^{p,S}$ and $\tilde{q}_i^{p,S}$ are defined for our specific problem setting when the (extended) set of base arms $\mathcal{B}$ consists of the union of two sets $\mathcal{Z}$ and $\mathcal{V}$. $q_i^{p,S}$ is the triggering probability of arm $a_i^z$ under action $S$ and is dependent on $p$ only. $\tilde{q}_i^{p,S}$ is the triggering probability of arm $a_i^v$ under action $S$. We did not write $\tilde{q}_i^{p,v, S}$ as the triggering probability is also dependent on $p$ only.
>
> P5 *The estimates of base arm outcomes are still unbiased?...*
>
> Yes, as explained in section 3.3, the estimates of base arms are biased and this is one of the key challenges we have for our problem. As a result, the Event E2 does not directly apply. To tackle with this difficulty, we introduce the concept of arm equivalence such that we can restore the CMAB-T framework by using some replacement items $(p',v')$. We note that $p_{i,t} '= p_iv_i, v_{i,t}' = 1$
>  when $v_i$ is unknown (first stage) and $p_{i,t} '= p_i, v_{i,t}' = v_i$ after observing $v_i$ (second stage). In this way, the estimates of base arms are unbiased in two stages. We then justify that using the replacement arms will not cause excessive regrets.
>
> P6 *Please comment on the batch-size dependence..*
>
> Yes. The current distribution-independent regret bound is $\tilde{O}(\sqrt{KT})$, the same as cascading bandits upper bound.
> $$R(T)\leq c_1\left(\sqrt{nKT} + \log\left(\sqrt{nKT} +1\right)\right)\log T +c_2 \left(\left(\log \left(\sqrt{nKT} +1\right)+1\right)\Delta_{\max}+1\right),$$
>
> P7 *What about batch size independent regret bounds as in Liu et al. 2022, NeurIPS...*
>
> Yes, we checked that our case satisfies the TPVM condition. We can improve the regret bound dependency from $k$ to $\log(k)$. The change is due to a tighter bound of the regret caused by estimation error of $p$. In the algorithm, besides maintaining a confidence bound for the mean value of $p$, we also maintain a confidence bound for its variance as Algorithm 1 in (Liu et al. 2022). We note that most of our methods and analysis remain the same. We give a summary of changes in the appendix.

---

### Official Review · Reviewer_AX9C · 2023-11-04

**Soundness:** 2 fair
**Presentation:** 3 good
**Contribution:** 2 fair
**Rating:** 6
**Confidence:** 3

**Summary:**

This paper studies the problem of combinatorial bandits with maximum value reward function with a intermediate feedback between semi-bandit and full-bandit. In particular, the learner chooses a subset $S_t \\subseteq [K]$ of $|S_t|=k$ arms as the action at each round $t$ and incurs the maximum realized reward $\\max_{i\\in S_t} X_t(i)$ among those arms, where $X_t(1),\\dots,X_t(K)$ are the i.i.d. stochastic rewards of the $K$ arms at round $t$. The learner then observes the incurred reward together with the identity $I_t \\in \\arg\\max_{i\\in S_t} X_t(i)$ of any arm achieving such a reward within the action. The authors consider discrete distributions with finite support (always including $0$) for the rewards and propose an algorithm achieving a regret bound for binary distributions that is comparable to previously-known bounds under more informative semi-bandit feedback. The results are first presented for the case of binary distributions with known ordering of arms with respect to the nonzero value in the support, and then generalized by first relaxing the assumption of a-priori known ordering. The paper also discusses extensions to discrete distributions with finite supports and presents numerical results that validate their theoretical findings.

**Strengths:**

- The paper considers a feedback structure for combinatorial bandits with maximum reward function that lies in between commonly studied semi-bandit and full-bandit feedback structures in combinatorial multiarmed bandit problems, recovering similar guarantees as the more informative semi-bandit setting for binary distributions.
- The authors present regret bounds for problem instances with stochastic arm outcomes according to binary distributions that always have $0$ in their support. The authors also argue how their techniques could generalize to the case of discrete distributions with finite support.
- The experimental results show that their proposed algorithm indeed achieves similar performance as an algorithm with semi-bandit feedback in this specific setting.
- The problem of combinatorial multiarmed bandits with maximum reward function can be relevant to many real-world scenarios.

**Weaknesses:**

- The paper could benefit from more detailed explanations of the proposed algorithm and its intuition. There is also no explicit definition of regret $R(T)$, which is a fundamental concept in the entirety of this work, especially for readers that might be unfamiliar with the combinatorial bandits literature. Its definition is only left implicit in how results are presented and proved, but the presentation could benefit from a formal definition in the problem formulation section. Only the "event-filtered regret" is defined, but its definition is deferred to the appendix. Finally, the authors keep referring to $R(T)$ as the regret, while its actually a more relaxed version known as $\\alpha$-regret (could be mentioned at least once in the problem setting).
- The paper focuses on binary distributions of arm outcomes always including $0$ in their support, which may not be representative of many real-world scenarios. The paper could benefit from exploring more general distributions of arm outcomes. Even if the authors argue about adapting to more general discrete distributions with finite support, these are also quite limited in the applications of their techniques. Furthermore, no theoretical guarantee is provided for there latter distributions, and the proposed reduction is expected to introduce a linear dependence in the support sizes to the regret. This is therefore unfeasible and undesirable for distributions with large enough supports, let alone countably infinite (or even continuous) supports as per many common distributions.
- The baseline methods used in the numerical results are never explicitly described or properly referenced. It is therefore unclear how meaningful the experimental results are without understanding what assumptions are made in the other algorithms used in the comparison.
- In line 7 of Algorithm 3.1 it says the learner is able to observe value $v_{i^*}$ of $i^*$ while it might happen that no value is observed if all arms in $S_t$ for a certain round $t$ have realized value equal to $0$.
- Some of the main ideas adopted in this paper, while adjusted to fit a setting with simpler distributions but less informative feedback, seem to be taken directly from Chen et al. (2016a) without striking changes.

References:
- Chen et al. (2016a): Wei Chen, Wei Hu, Fu Li, Jian Li, Yu Liu, and Pinyan Lu. "Combinatorial multi-armed bandit with general reward functions". *Advances in Neural Information Processing Systems*, 29, 2016.

**Questions:**

- Is the presence of $0$ in the support strictly necessary? Or do you believe it is possible to lift this assumption? If so, could you hint at the intuition for doing so?
- Please, see weaknesses and further elaborate on those points.

---

> ### Author Response · Authors · 2023-11-20
> **Thanks for the review. We have addressed the questions asked.**
>
> We thank the reviewer for positively appreciating the presentation of our work and providing us with detailed comments. However, we think the reviewer misunderstood some parts of our work and we do not agree with some of the comments that qualify our technical contributions to be a straightforward modification of existing results. We give detailed explanations of the questions asked in the following. More updates are included in the appendix, and we will incorporate all addressed points in the final version of our paper.
>
> P1 *The paper could benefit from more detailed explanations of the proposed algorithm and its intuition. There is also no explicit definition of regret, which is a fundamental concept in the entirety of this work, especially for readers that might be unfamiliar with the combinatorial bandits literature.*
>
> We are sorry that we mistakenly removed a paragraph in the problem formulation section. We add back the formal definition of the regret and the approximation oracle as follows.
>
> P2 *The paper focuses on binary distributions of arm outcomes always including $0$ in their support.. Even if the authors argue about adapting to more general discrete distributions..*
> P3 *Is the presence of $0$ in the support strictly necessary?*
>
> No, it is totally fine to have arms without 0 in its support. We did not impose this constraints for arm distributions. Please take another look at section 2 problem formulation on page 3. We require that $0 < \sum_{j=1}^{s_i} p_{i,j}\leq 1$. Note that when the right inequality holds with equal sign, $p_{i,0} = 0$ and the arm outcome does not include zero in the support. In the binary case, this simply means that we have a deterministic arm. In the general case, the equivalence relationship is not affected. For example, if $X_i$ takes value $0.1$ with probability $2/3$, and value $0.22$ with probability $1/3$, then by eqn.(4.1), we can represent $X_i$ by two random variables $X_{i,1}$ and $X_{i,2}$ such that $X_{i,1}$ takes value $0.1$ with probability $1$, and $X_{i,2}$ takes value $0.2$ with probability $1/3$ and value 0 with probability $2/3$. One can easily verify that $Y=\max\{X_{i,1}, X_{i,2}\}$ follows the distribution exactly the same as the original $X_i$, namely with probability $2/3$ $Y=0.1$ and with probability $1/3$ $Y=0.2$.
>
> We also argue that our setting is not limited in the applications, as categorical ratings or ranks are commonly used in the real world. The general algorithm on discrete distributions with finite support does have theoretical guarantees. As we discussed in the main text, equipped with the equivalence relationship, we are able to treat arms with outcomes according to arbitrary distributions with finite supports as a set of arms with outcomes according to binary distributions and use our algorithm defined for the case of arms with binary distributions. Specifically, we have the following claim for regret upper bound,
>
> For the $k$-MAX problem with max value-index feedback, under general case of arbitrary discrete distributions of arm outcomes with finitesupport as defined in Section 2 and under assumption $\Delta_{\min}>0$, Algorithm 4.1 has the following distribution-dependent regret bound,
> $$R(T)\leq c_1 \sum_{i=1}^n\left(\frac{k}{\Delta_{\min}^i} + \log\left(\frac{k}{\Delta_{\min}^i}+1\right)\right) s_i\log(T) +c_2 \sum_{i=1}^n s_i\left(\left(\log \left(\frac{ k}{\Delta_{\min}^i}+1\right)+1\right)\Delta_{\max}+1\right)$$
> We give further comments of the claim in the appendix.
>
> We acknowledge that our current regret bound does depend on the support size of the discrete variables. We believe this does not limit the applicable scenarios of our algorithm. In many applications, the support size is relatively small. For application scenario where the support size is large, we would like to emphasize the distinction between the choices that the platform offers and the internal values of the items that a user actually perceives. In may cases, even though the support size may be infinite from the values offered by the platform (e.g. real-valued feedback), the actual value of the customer on an item may only have a few possible internal values.
>
> We note the possibility of extending our result to allow for continuous reward distributions by discretizing continuous arm distributions. We do like to mention that when all the values of the variables are unknown, intuitively the exploration does seem to need to explore all possible support values of variables in order to exploit the best option, and this may suggest some inherent dependency on the support size or a tradeoff between discretization granularity and regret bound. This is indeed an intriguing research direction for future research	to study the upper and lower bound dependencies on the support size, but we believe it is out of the scope of the current paper, while our current paper already has proper technical innovations that cover a reasonable range of practical applications.

---

> > ### Author Response · Authors · 2023-11-20
> > **Question addressed (cont.)**
> >
> > P1 (cont. Formal definition of regret)
> > The performance of the agent is measured by the cumulative regret, defined as the difference of the expected cumulative reward achieved by playing the best action and the expected cumulative reward achieved by the agent. Denote $\mathrm{OPT}(p,v) = \max(\{r_S(p,v) \mid S\in \mathcal{F}\})$. An $(\alpha,\beta)$-approximation oracle takes  $(p',v')$ as input and returns a set $S$ such that
> > $
> > \Pr[r_S(p',v')\geq \alpha\ \mathrm{OPT}(p',v') ]\geq \beta
> > $
> > where $\alpha$ is the approximation ratio and $\beta$ is the success probability.
> > If the agent uses an $(\alpha,\beta)$-approximation oracle, then we consider the $(\alpha,\beta)$-approximation regret defined as
> > $$
> > R(T) = T\ \alpha\ \beta\ \mathrm{OPT}(p,v) - E\left[\sum_{t=1}^{T} r_{S_t}(p,v)\right].
> > $$
> >
> >
> > P4 *The baseline methods used in the numerical results are never explicitly described or properly referenced.*
> >
> > We will explicitly cite the work for the baseline methods in the numerical section. Currently, we referred to the standard CUCB method at the beginning of section 2 literature review and in the appendix A. It is the previous work on the $k$-MAX bandit problem with semi-bandit feedback.
> >
> > P5 *In line 7 of Algorithm 3.1 it says the learner is able to observe value of while it might happen that no value is observed.*
> >
> > Yes, this is possible. In this case, there is no winner and we do not update $v$. We will comment that we update $v$ values only when the winner value is non-zero.
> >
> > P6 *Some of the main ideas adopted in this paper, while adjusted to fit a setting with simpler distributions but less informative feedback, seem to be taken directly from Chen et al. (2016a) without striking changes.*
> >
> > This is one of the key reference papers. We take the offline oracles from Chen et al. (2016a); however, we argue that the assumptions, key ideas and methods of our paper are significantly different from Chen et al. (2016a). Our main challenge stems from the fact that Chen et al. (2016a) studied semi-bandit feedback while our problem has max value-index feedback. Unlike Chen et al. (2016a), our feedback is much weaker and we are not able to apply the SDCB algorithm that requires maintaining empirical distributions of the observed arm outcomes. We developed a variant of the CUCB algorithm to solve our problem. We established an equivalence relationship between general distributions of finite support and binary distributions. Our analysis rests on a novel idea of considering an extended set of base arms, associated with $v_1,\ldots, v_n$ and $p_1,\ldots, p_n$ values. The unknown values $v_1,\ldots,v_n$ cause significant technical challenges for our problem which we circumvent in our analysis. In the case of unknown value ordering, we cannot infer whether a base arm is triggered or not based on the winner value and the winner index only. We tackle this difficulty by introducing the concept of item equivalence. These are all new technical contributions to address issues that arise because of the different types of feedback we study.

---

### Official Review · Reviewer_hUTf · 2023-11-05

**Soundness:** 3 good
**Presentation:** 2 fair
**Contribution:** 2 fair
**Rating:** 6
**Confidence:** 3

**Summary:**

The authors consider MAB problems where the action set is combinatorial (subsets with cardinality k) and the reward function is the max reward over the base arms.  Base arms are discrete valued and the authors consider both known and unknown support.  The authors propose a variant of semi-bandit feedback where the agent (only) learns which base arm had the largest value.   The authors then adapt the CUCB algorithm for this setting and obtain problem independent regret bound dependence of $\tilde{O}(\sqrt{T})$.

**Strengths:**

### Strengths

- Novelty – There is some novelty in the problem, method, and analysis.  K-max has been studied under semi-bandit feedback (all base arm values revealed; Chen et al. (2016)) and for special cases (Bernoulli arms) studied under bandit feedback (see note about (Agarwal et al. (2021)) below).  The proposed max-index feedback considered shares some similarity with other feedback models (cascading, triggering, dueling, etc) but I am not aware of this specific feedback model being studied previously.  The algorithm and analysis largely (though not trivially) follow that for the CUCB for probabilistically triggered arms -- the realization and index of the maximizing base arm are modeled as two arms that are triggered.
- Significance – I think with stronger motivation for the problem set up considered this would be of some interest to the CMAB community.
- Quality – I did not go through the proofs, but from my reading the work appears sound.

**Weaknesses:**

## Weaknesses
### Related work
- There’s an uncited work that is relevant -- https://ojs.aaai.org/index.php/AAAI/article/view/16812 “DART” by the same authors of (Agarwal et al. (2021)), for picking K sized subsets with a non-linear rewards.  The former does not have the same FSD assumption as the cited work and in terms of horizon $T$ dependence they get $\tilde{O}(\sqrt{T})$ problem independent upper regret bounds and lower bounds.  They show that K-max for independent Bernoulli arms satisfies the assumptions in that paper.



### Writing & Clarity
There are a few (addressable) issues with the writing that affect readability.

- Is the regret $R(T)$ formally defined?  I didn’t see it in the problem formulation or anywhere before it is upper bounded in Theorem 3.4.  In addition to being unambiguous about cumulative vs instantaneous, regret vs pseudo-regret, you mentioned approximation algorithms for the general offline problem and so it becomes ambiguous if you are working with regret or an $\alpha$-regret (and if so specifically a $\alpha=1-1/e$ regret or an $\alpha=1-\varepsilon$ regret).

- Relatedly, page 5 is the first place “approximation oracle” is mentioned once in the text before the Alg 3.1 pseudocode but what “approximation oracle” refers to is not formally described.  That should be formally discussed in the problem setup or at least before pseudocode for the algorithm that calls it.  Also, for completeness the procedures that are proposed to be used for an oracle should be formally described (appendix would be fine).


- Algorithms –
    - (minor) I found the discussion in the text confusing about maintaining confidence bounds on not only $p_i$’s but also $v_i$’s, which are deterministic.  Looking in the pseudocode for the $v_i$’s there aren’t any confidence bounds – instead the values are set optimistically to $1$ and (since the support is deterministic) updated exactly once on the first time that arm is the maximizer (if that occurs).  Calling it a confidence bound is confusing in my opinion.
    - (Minor) – In Section 3.2 I would suggest linking to the pseudocode earlier, such as at end of second paragraph “Due to limited space, the pseudocode is presented in Algorithm B.1.”

- I think it would be valuable to have a fuller discussion of the hardness for the offline setting and corresponding approximation methods (Bernoulli (which seems easy based on (Agarwal et al. (2021))’s DART results), non-Bernoulli binary (you mention can be solved exactly with dynamic programming, but are hardness results known for the non-Bernoulli binary case?), and general (you cite Chen et al. (2016a).) to give the reader a fuller intuition.  E.g. explicitly state that the general offline problem (and non-Bernoulli binary offline problem too) is NP-hard; don’t just state that greedy methods can get an $\alpha$ approximation.


### Experiments
- Semi-bandit CUCB is good to include; it would be good to also include Agarwal et al. (2021))’s DART since that has  $\tilde{O}(\sqrt{T})$ guarantees under bandit feedback when the arms are Bernoulli.  It would probably do worse since its assumptions might not hold for non-Bernoulli arms, but it might be more competitive than UCB.
- The distributions in the experiments are all binary?  That should be mentioned in the main text.
- “three different arm distributions representing different scenarios” is quite generic and suggests potentially distinct problems (such as modeling three different applications), but the ‘different scenarios’ are almost identical manually constructed examples with a single arm changed from scenarios 1 to 2 and then from 2 to 3.  That is ok to do but the description in Section 5 should be revised to reflect both of those issues (manually designed and single arm changes from one scenario to another).

**Questions:**

1. Motivation – What would be a motivating application of K-max with the max-index feedback for which the arm supports would be unknown? Elaborate more on how one or multiple applications motivate the problem set up you consider.  The ‘concrete’ scenario mentioned in the introduction of recommender systems “The goal is to display a subset of k product items to a user that best fit the user preference. The feedback can be limited, as we may only observe on which displayed product item the user clicked and their subsequent rating of this product.”  For most systems I can think of, the numerical feedback would be on a known, fixed scale (like 1-5 stars, or on a 1-10 scale) meaning the support is identical for all arms and known a priori.



### Minor
2. In the contributions, “Our work may be seen as a step towards solving full-bandit CMAB problems with non-linear reward functions under mild assumptions.”  Can you elaborate on that (if that's kept in the main contributions)?  The feedback model and modifications to CUCB to appear specialized for the max reward function and thus the work does not seem extendable to other non-linear rewards.

---

> ### Author Response · Authors · 2023-11-20
> **Thanks for the detailed comments!**
>
> We would like to thank the reviewer for positively appreciating the technical soundness and novelty of the problem we study, and providing detailed comments on the paper presentation. More updates are included in the appendix, and we will incorporate all addressed points in the final version of our paper.
>
> P1 *here’s an uncited work that is relevant -- “DART” by the same authors of (Agarwal et al. (2021).*
>
> Thanks for pointing this out. We will cite this work in the related work section on full-bandit CMAB problems. We note that our $k$-MAX bandit setting does not satisfy the assumption 2 Good arms generate good actions in Agarwal et al. (2021). High-risk high reward item in our setting could be favored by the maximum reward. Consider $n=3$ and $k=2$ such that $X_1$ taking 0.9 with probability 0.2, $X_2$ taking 0.5 with probability 0.4 and $X_3$ taking 0.6 with probability 1. In this case, $E(X_1)<E(X_2)$ but $X_1$ is in the optimal set instead of $X_2$.
>
>
> P2 *Is the regret $R(T)$ formally defined... Relatedly, page 5 is the first place approximation oracle is mentioned once in the text before the Alg 3.1 pseudocode but what “approximation oracle” refers to is not formally described.?*
>
> We truly apologize for the ambiguity caused. We have mistakenly removed one paragraph from the problem formulation section that formally defines the regret and the approximation oracle with the procedures. It is added back as a section in the appendix.
>
> P3 *Algorithms I found the discussion in the text confusing about maintaining confidence bounds..*
> P4 *"In Section 3.2 I would suggest linking to the pseudocode earlier.*
>
> The confidence radius of $v_i$ is defined as $\boldsymbol{1}\{\tilde{T}_i = 0\}$. It is defined in this way for notation convenience and consistency. Note that the value may never be revealed. We will make it explicit in Algorithm B.1 and link the pseudocode at the end of second paragraph of section 3.2.
>
> P5 *I think it would be valuable to have a fuller discussion of the hardness for the offline setting and corresponding approximation methods.*
>
> Thanks for the good question. We give a full discussion of the hardness results. For the general offline $k$-MAX problem (Goel et al 2014), finding the exact optimal solution is NP-hard. (Chen et al 2016) shows that there exists a PTAS for the offline $k$-MAX problem such that for any constant $\epsilon>0$ there is a polynomial-time $(1-\epsilon)$-approximation algorithm for the offline $k$-MAX problem. The binary and Bernoulli $k$-MAX problems are not NP-hard. The binary $k$-MAX problem can be solved exactly using dynamic programming. The Bernoulli $k$-MAX problem can be solved exactly by ordering values and selecting the top $k$.
>
> P6 *Semi-bandit CUCB is good to include; it would be good to also include Agarwal et al. (2021))’s DART.*
>
> We will test the performance of the DART algorithm. We note that the current experimental setup is designed purposely to show that the algorithm works for scenarios with high-risk high-reward items or stable-valued items with low reward. As mentioned, this assumption does not hold in Agarwal et al. (2021)).
>
> P7 *The distributions in the experiments are all binary?*
> P8 *"three different arm distributions representing different scenarios" is quite generic..*
>
> Thanks for pointing these out. We will clarify the experimental setup as suggested.
>
> P9 *Motivation – What would be a motivating application of K-max with the max-index feedback for which the arm supports would be unknown? *
>
> Firstly, we argue that in some cases, the agent has no knowledge of the feedback scale. For example, in bidding or donation, the agent would have no access to the internal value of the bidders or donors beforehand; in sensor detection, the agent does not know the informativeness of sensors before receiving signals.
>
> Moreover, we would like to emphasize the distinction between the choices that the platform offers and the internal values of the items that a user actually perceives. On most systems, we may offer choices on a 1-10 or 1-100 scale. If we treat this as the case of known support size, the support size would be 10 or 100, respectively. However, the user/customer may value a given item with only one or a few possible values. By treating these possible values as unknown supports, we may have a much smaller support size to deal with on each random variable, and thus resulting into smaller regret.
>
> P10 *Our work may be seen as a step towards solving full-bandit CMAB problems..?*
>
> In our current work, we consider the expected maximum as the reward function. We acknowledge that the proof techniques are tailored to leverage properties of the expected maximum. We note that the idea of arm equivalence and using fictitious arms can be possibly applied to solve CMAB problems of other non-linear rewards. We will downplay a bit of this point of contribution.

---

> > ### Comment · Reviewer_hUTf · 2023-12-05
> >
> > I thank the authors for their response and clarifications.

---

### Meta-Review · Area_Chair_DJWK · 2023-12-05

**Metareview:**

The authors address a combinatorial multi-armed bandit problem within the context of a maximum value reward function, incorporating both maximum value and index feedback. This unique feedback structure falls between the extensively investigated semi-bandit and full-bandit feedback structures. An algorithm is introduced, accompanied by a regret bound for instances of the problem where arm outcomes follow stochastic processes with arbitrary distributions and finite supports. The regret analysis is grounded in the consideration of an extended set of arms, each associated with values and probabilities of arm outcomes, while leveraging a smoothness condition.

Strengths:
1. New feedback structure
2. Analysis of regret bounds
3. Evaluations

Weaknesses:
1.  lower bound is not provided.
2. Some baselines have been mentioned in reviews and author rebuttal that would be good to compare in the revision.
3. The reviewer point out that the results of Katariya et al. (2017) summarized in Table 1 do not seem to make the assumption of Bernoulli rewards, as they work with stochastic rewards in [0,1]—see Section 2 therein. This should be fixed in the revision.

**Justification For Why Not Higher Score:**

The paper will be of interest to a smaller community of ICLR. The reviewers express a somewhat tepid positive reception to the paper.

**Justification For Why Not Lower Score:**

The paper has garnered positive response from all reviewers, and the rebuttal effectively addressed the key concerns raised.

---

### Decision · Program_Chairs · 2024-01-16

Accept (poster)